# *Candida albicans* evades NK cell elimination via binding of Agglutinin-Like Sequence proteins to the checkpoint receptor TIGIT

Yoav Charpak-Amikam [1], Tom Lapidus[1], Batya Isaacson[1], Alexandra Duev-Cohen[1], Tal Levinson [2], Adi Elbaz[3], Francesca Levi-Schaffer [4], Nir Osherov[5], Gilad Bachrach[6], Lois L. Hoyer[7], Maya Korem[8], Ronen Ben-Ami[2] & Ofer Mandelboim [1✉]

*Candida albicans* is the most common fungal pathogen and a prevalent cause of deadly bloodstream infections. Better understanding of the immune response against it, and the ways by which it evades immunity, are crucial for developing new therapeutics against it. Natural Killer (NK) cells are innate lymphocytes best known for their role against viruses and tumors. In recent years it became clear that NK cells also play an important role in anti-fungal immunity. Here we show that while NK cells recognize and eliminate *C. albicans*, the fungal cells inhibit NK cells by manipulating the immune checkpoint receptor TIGIT (T cell immunoreceptor with Ig and ITIM domains) in both humans and mice. We identify the responsible fungal ligands as members of the Als (Agglutinin-Like Sequences) protein family. Furthermore, we show that blocking this interaction using immunotherapy with a TIGIT-blocking antibody can re-establish anti-*Candida* immunity and serve as a potential therapeutic tool.

---

[1] The Concern Foundation Laboratories at the Lautenberg Center for Immunology and Cancer Research, Hebrew University Medical School, IMRIC, Jerusalem 91120, Israel. [2] Infectious Diseases Unit, Tel Aviv Sourasky Medical Center and the Sackler Faculty of Medicine, Tel Aviv University, Tel Aviv 64239, Israel. [3] Department of Microbiology and Molecular Genetics, Institute for Medical Research Israel-Canada (IMRIC), Faculty of Medicine, Hebrew University of Jerusalem, Jerusalem, Israel. [4] Pharmacology and Experimental Therapeutics Unit, School of Pharmacy, Institute for Drug Research, Faculty of Medicine, Hebrew University of Jerusalem, Jerusalem, Israel. [5] Department of Clinical Microbiology and Immunology, Sackler School of Medicine, Tel-Aviv University, Ramat-Aviv, Tel-Aviv 69978, Israel. [6] The Institute of Dental Sciences, The Hebrew University-Hadassah School of Dental Medicine, Jerusalem, Israel. [7] Department of Pathobiology, University of Illinois at Urbana-Champaign, Urbana, IL, USA. [8] Department of Clinical Microbiology and Infectious Diseases, Hadassah-Hebrew University Medical Center, Jerusalem, Israel. ✉email: oferm@ekmd.huji.ac.il

Candida albicans is a major fungal component of the human microbiome in most people[1,2]. It is highly active immunologically and recent studies have established a role for it in training and priming the immune response under healthy physiological conditions[3–6]. Certain triggers can drive the fungus to change its lifestyle and turn pathogenic, and indeed, C. albicans is the most common fungal pathogen of humans[2]. This pathogenicity is manifested by a range of diseases. The most common is vulvovaginal candidiasis, an infection of the mucosal surfaces of the female genital tract. Three quarters of all women suffer from at least one infection during their lifetime, and up to 8% of women suffer from recurrent infections[7]. The most lethal form of candidiasis is invasive candidiasis, a condition that commonly begins as a dysbiosis of the gut microbiome, but quickly deteriorates as C. albicans manages to breach the mucosal barrier and invade the host bloodstream and internal organs[1,8]. This form of infection is the 3rd to 4th most common hospital acquired bloodstream infection in many developed countries, accounting for 18–22% of such infections in the US, and leading to death in up to half the cases[1]. The recent rise in infections by Candida species harboring multi-drug resistances[1,9] is another worrying complication.

Both innate and adaptive immune cells are active against fungal infections[3]. Indeed, congenital and acquired deficiencies in either Natural Killer (NK) or T cells lead to increased rate and severity of various fungal infections[3,10,11]. While the role of T cells, and specifically $T_H1$ and $T_H17$ cells in control of fungal infections is better understood, the function of NK cells against fungal infections is far less clear. NK cells perform their immune activities using three main pathways; de-granulation and secretion of lytic and pro-apoptotic proteins (such as perforin, granulysin, or granzymes), induction of apoptosis via death receptor signaling (such as FAS and TRAILR), and recruitment of additional immune cells by secretion of pro-inflammatory cytokines and chemokines (such as IFNγ, TNF-α or GM-CSF)[12,13]. Murine NK cells have been shown to confer their anti-fungal protection using two of these mechanisms; either directly by de-granulation and killing of C. albicans cells, or indirectly by secretion of cytokines and chemokines such as IFN-γ and GM-CSF[14–17]. However, the protective role of NK cells in bloodstream C. albicans infections of various mouse models is still under debate, with some previous works showing a protective role[15], while others present a more complicated scenario in which NK cells can be either protective or detrimental under different conditions[18]. As expected, the data regarding the involvement of NK cells in human systemic Candida infections are more limited, but both in-vitro data and at least one prospective non-interventional clinical study demonstrate a potential anti-fungal role for NK cells[14,19,20].

NK cells recognize their targets using an array of activating receptors and inhibitory immune checkpoint receptors. Several activating receptors have been identified as fungal sensors[1,3,17,21]. Recently the NK-receptor NKp30 has been shown to recognize the C. albicans cell wall sugar β-glucan[22,23], and its family member NKp46 was shown to recognize another Candida species, C. glabrata, through members of the Epa fungal cell wall adhesion proteins[24]. It is yet unknown whether any inhibitory immune checkpoint receptors recognize fungi in general and C. albicans in particular, and what could be the outcome of such recognition.

T-cell immunoglobulin and immunoreceptor tyrosine-based inhibitory motif (ITIM) domain (TIGIT) is one such example of an inhibitory immune checkpoint receptor[25–27]. It is a surface receptor, expressed on NK cells and several T cell subtypes[28]. In humans, it binds members of the Nectin family of proteins such as PVR, Nectin2, Nectin3, and Nectin4[26,29,30]. TIGIT activation leads to inhibition of the immune response, either directly by inhibiting TIGIT-expressing pro-inflammatory cells, or indirectly by activating specific anti-inflammatory cells, such as $T_{reg}$ cells, and secretion of anti-inflammatory cytokines such as IL-10[28]. It is hypothesized that the physiological role of TIGIT is the timely resolution of immune responses and prevention of immunopathology, but it is also known to malfunction during malignant processes and take part in lymphocyte exhaustion[28,31–33]. In one such case, TIGIT was shown to be manipulated in patients suffering from colorectal cancer for the benefit of the tumor and its associated bacterial species, Fusobacterium nucleatum. In the described case, the Fap2 protein of F. nucleatum was shown to bind TIGIT and inactivate anti-tumor lymphocytes, providing immune evasion for the spreading cancer[34–36].

Major progress in cancer therapy was recently achieved via the use of antibodies that block immune checkpoint receptors such as CTLA-4 and PD-1 or the PD-1 ligand PD-L1[37]. TIGIT is currently also being evaluated as an immune checkpoint target for the treatment of cancer[37–39]. It was suggested that similar checkpoint therapy can be used during chronic infections[40], and proof-of-concept studies have also been performed on models of several fungal infections[41–44]. Such treatments can be especially useful in the case of high-mortality fungal infections for which we do not have sufficient treatments, as in the case of invasive candidiasis. Up until now no fungal ligand of TIGIT is known, and the role of TIGIT in fungal infections was hypothesized to only be the cessation of the immune response and the prevention of immunopathology.

In the current study, we show that a human fungal pathogen, C. albicans, can bind and activate the immune checkpoint receptor TIGIT on NK and T cells in both humans and mice as a mechanism for immune evasion. We identify the responsible fungal ligands; proteins belonging to the Agglutinin-Like Sequences (Als) family of adhesins. Finally, we demonstrate that interfering with this interaction using a specific TIGIT-blocking antibody improves host outcomes in vivo, suggesting a possible future therapeutic potential in C. albicans infections.

## Results

**TIGIT directly binds C. albicans yeast and hyphae cells**. To test how NK cells recognize C. albicans we generated a library of recombinant fusion proteins containing the extracellular ligand-binding domain of known human NK receptors fused to the Fc domain of human $IgG_1$. Using flow cytometry, we analyzed the binding of the various NK receptors to C. albicans yeast cells. As shown, binding was observed between the fusion protein TIGIT-Ig and the yeast cells (Fig. 1a, quantified in B). To corroborate this finding, we bound the cells to 96-well plates and used ELISA assays with TIGIT-Ig and again observe binding (Fig. 1c).

To test whether TIGIT binding is C. albicans-specific, we stained three additional pathogenic Candida species; C. glabrata, C. parapsilosis and C. krusei, and two clinical isolates of non-albicans pathogenic fungi; Cr. neoformans and Cr. gattii, with TIGIT-Ig. Of all examined fungi, only C. parapsilosis was significantly stained with TIGIT-Ig (Fig. 1d).

We decided to focus here on C. albicans as it is the most common cause of fungal infections in humans[2], and as it is relatively more studied and better understood in comparison to other members of the Candida genus.

C. albicans is a multi-morphic fungus that can differentiate into several cell types, the most notable examples being the round yeast cells and the elongated hyphal cells. Hyphal C. albicans cells are critical for their pathogenicity and invasive potential[2,45], and we therefore wished to check whether hyphal cells might also be

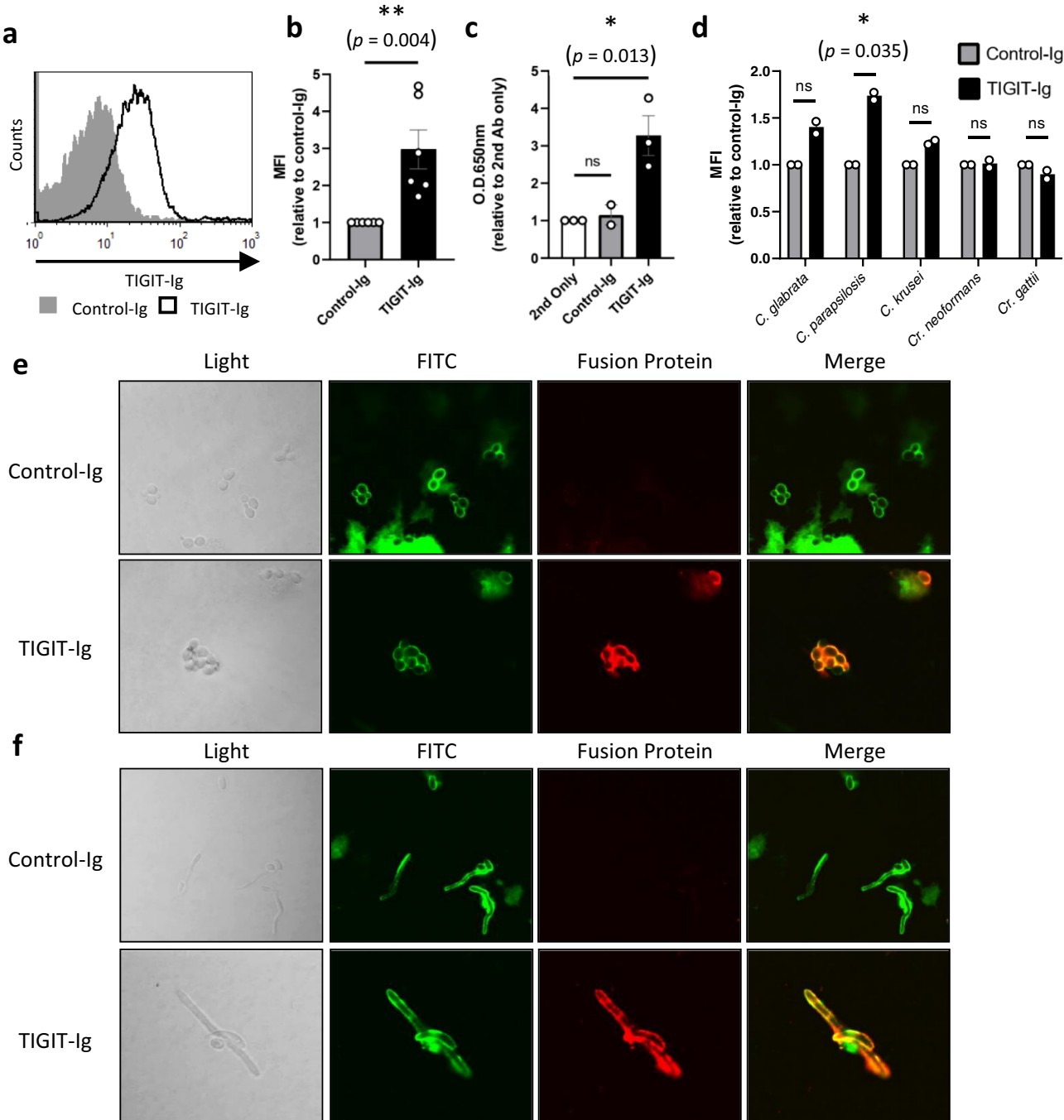

**Fig. 1 TIGIT directly binds *C. albicans* yeast and hyphae cells. a** Flow cytometry staining of *C. albicans* SC5314 yeast cells using TIGIT-Ig (Black empty histogram) or a negative control protein (filled gray histogram). One representative experiment out of 6 is presented. **b** Quantification of the results presented in (**a**). $n = 6$ independent experiments. **c** ELISA results of plate-bound *C. albicans* cells stained with TIGIT-Ig or a negative control protein. $n = 3$ independent experiments. **d** Quantification of flow cytometry stainings of various fungal species using TIGIT-Ig or a negative control protein. $n = 2$ independent experiments. **e** Immunofluorescence microscopy images of *C. albicans* yeast cells stained for their cell wall proteins with FITC (green) and either TIGIT-Ig or a negative control protein (red). **f** Immunofluorescence microscopy images of *C. albicans* hyphae cells stained for their cell wall proteins with FITC (green) and either TIGIT-Ig or a negative control protein (red). For figures (**e**, **f**) representative images from two independent experiments are presented. Data for (**b–d**) are presented as mean values ± SEM. Significance was tested using a two-tailed Student's *t* test. ns = not-significant, * = $p < 0.05$, ** = $p < 0.01$.

recognized by TIGIT as we observed for the yeast cells. As it is technically problematic to use hyphae in flow cytometry and ELISA experiments, we instead used our TIGIT-Ig-fusion protein to stain yeast and hyphal cells using confocal microscopy. In agreement with the above results, we observed staining of yeast cells with TIGIT-Ig, but not with a negative control fusion protein (Fig. 1e). Importantly, we also observed TIGIT-Ig binding to *C. albicans* hyphae (Fig. 1f).

Thus, we concluded that the human TIGIT receptor binds the two major *C. albicans* morphotypes.

***C. albicans* binding of TIGIT is functional and leads to inhibition of NK and T cells**. To test whether TIGIT recognition of *C. albicans* is functional, and to directly assess the TIGIT- *C. albicans* interaction, we expressed TIGIT in the NK cell line YTS Eco which does not endogenously express TIGIT (Supplementary Fig. 1A, B). The parental cell line (YTS Eco) and the TIGIT-expressing cell line (YTS TIGIT) were co-incubated with *C. albicans* cells in the presence or absence of a monoclonal α-human TIGIT antibody. This antibody has been shown to block the interaction between TIGIT and its endogenous human ligands[25] and we hoped it will also block similar interactions with possible fungal ligands. After 12 h of co-incubation the cells were serially diluted, plated on Sabouraud agar plates, and the resulting fungal colonies were counted on the following day. Percent Colony Forming Units (CFU) reduction was calculated by comparing the resulting colonies to a control experiment performed in the absence of YTS cells. As YTS TIGIT originates from a single clone of YTS Eco cells, the two lines have different expression levels of some proteins, including some that might be relevant for their anti-fungal activity (unpublished data). This could lead to different basal levels of anti-candida cytotoxicity. Therefore, we only compared TIGIT-blocked and isotype-controlled samples within each cell line, and not between the different lines.

TIGIT blockade significantly increased the ability of YTS TIGIT cells to eliminate *C. albicans* cells but had no effect on the parental YTS Eco cells (Fig. 2a). We next tested the validity of our observations using primary human NK cells. We isolated NK cells from the blood of healthy human donors, activated them, and co-incubated them with *C. albicans* cells in the presence or absence of the TIGIT-blocking antibody. Remarkably, TIGIT blockade significantly increased the elimination of fungal cells by primary NK cells (Fig. 2b). This led us to conclude that the above-mentioned antibody can block the interaction between TIGIT and its fungal ligands, and that *C. albicans* activation of TIGIT is functional and inhibitory for NK cells.

As TIGIT is expressed on both NK and T cells, we next examined whether TIGIT activation by *C. albicans* can also inhibit T cells. We isolated primary CD4+TIGIT+ T cells from healthy human donors and cultured them under conditions that favor differentiation into $T_H1$ cells. These T cells were then activated using a αCD3 antibody bound to the Fc receptor expressed on the murine P815 mastocytoma cell line. This leads to TCR crosslinking, T cell activation, and IFNγ secretion (Fig. 2c). We expected that addition of the anti-TIGIT-blocking antibody would enhance T cell activity, but only in the presence of a TIGIT-stimulating signal, in this case, *C. albicans* (Fig. 2c). Indeed, TIGIT blockade in the absence of *C. albicans* did not lead to increased IFNγ secretion, but in the presence of *C. albicans* we observed a significant elevation in IFNγ secretion (Fig. 2d).

We next wished to examine whether the signal produced by *C. albicans* activation of TIGIT is transduced into the cell using the same pathways as the those produced by endogenous human TIGIT ligands. As was previously shown, a critical initial signaling step during activation of TIGIT is phosphorylation of its ITIM domain which also gives the protein its name (T cell immunoreceptor with Ig and ITIM domains). Mutating the critical tyrosine residue within this motif (Y231) abrogates classical TIGIT signaling and the downstream inhibitory effects[25]. We have previously generated a YTS cell line expressing Human TIGIT mutated in this critical Y231 residue; YTS TIGIT Y231A. We also generated another cell line, YTS TIGIT Y231Stop, in which we introduced a stop mutation in the ITIM domain and eliminated the final 13 residues of the protein. After validating TIGIT expression in these cell lines (Supplementary Fig. 1C, D) we co-incubated them with *C. albicans* cells in the presence or

absence of the TIGIT-blocking antibody and quantified their antifungal capacity and their response to TIGIT blockade as described in Fig. 2a, b. TIGIT blockade was only able to significantly increase the antifungal capacity of YTS cells expressing the wildtype TIGIT gene and had no effect on cells expressing TIGIT mutated in the ITIM motif (Fig. 2e). This suggests that the signaling cascade initiated by TIGIT binding to its *C. albicans* ligands is mediated by its ITIM domain, similar to the canonical signaling pathway activated upon its binding of the endogenous human ligands.

In order to the find whether this phenomenon might play a role in human clinical disease we collected clinical isolates of *C. albicans* from a cohort of 105 human patients who suffered from *C. albicans* bloodstream infections. The ability of these clinical isolates to manipulate TIGIT was determined using a BW assay. The BW assay is a semi-functional screen in which the extracellular portion of human TIGIT is fused to a murine ζ-chain and expressed in the murine thymoma cell line BW. Functional binding of a ligand to TIGIT leads to activation of the BW cells and secretion of IL-2. BW-TIGIT cells (Supplementary Fig. 1E, F) were incubated in the presence or absence of fluconazole-inactivated *C. albicans* cells for 48 h and IL-2 levels were measured using ELISA. We measured the TIGIT activation ability of each clinical strain and compared it to the WT lab strain SC5314. Importantly, all clinical strains activated TIGIT, as they all activate BW-TIGIT cells at least to the same extent as the WT lab strain (Fig. 2f).

Thus, we concluded that *C. albicans* recognition by TIGIT is a functional immune evasion mechanism that enables the fungus to inhibit NK cells and T cells in-vitro and is also clinically relevant.

**Als proteins of *C. albicans* are fungal TIGIT ligands**. We next set out to identify the fungal ligands of TIGIT. We started by scanning a published library of *C. albicans* deletion mutants, encompassing ~11% of the *C. albicans* genome[46] and screened these mutants in an ELISA assay as described in Materials and Methods. We used an ELISA assay for this screen since it enables parallel screening of a large number of fungal strains. However, no mutant was found to significantly alter TIGIT-Ig binding.

We next proceeded to test additional *C. albicans* proteins not represented in the deletion library. Microbial NK ligands such as the Epa proteins of the fungus *C. glabrata*[24], the Fap2 protein of the bacterium *F. nucleatum*[34], and viral proteins such as the influenza virus protein HA[47], share several characteristics. These include being important pathogenicity factors, functioning as adhesion molecules, and localizing to the cell surface of the pathogen or the cells it infects. In addition, many NK-receptor ligands belong to the Ig-superfamily of proteins. An important *C. albicans* protein family that shares all these characteristics and is not represented in the deletion library we scanned (other than a single member, Als5) is the Agglutinin-Like Sequences (Als) family. The Als protein family consists of 8 members of cell-wall-bound proteins used mainly for adhesion to various biotic and a-biotic surfaces, and is central for the pathogenicity of *C. albicans*[48,49].

In order to check whether Als proteins could serve as TIGIT ligands we performed a BW assay using BW-TIGIT cells with a series of *C. albicans* mutants, each deleted for both alleles of a member of the Als protein family. We compared the ability of each mutant to activate TIGIT relative to the WT strain and identified three Als mutants impaired in TIGIT activation; *als6Δ/Δ*, *als7Δ/Δ*, and *als9Δ/Δ* (Fig. 3a). The importance of these genes in TIGIT-mediated immune evasion was then examined using primary human NK cells used in a cytotoxicity assay. Only the WT strain of *C. albicans* was sensitive to TIGIT blockade, while the deletion

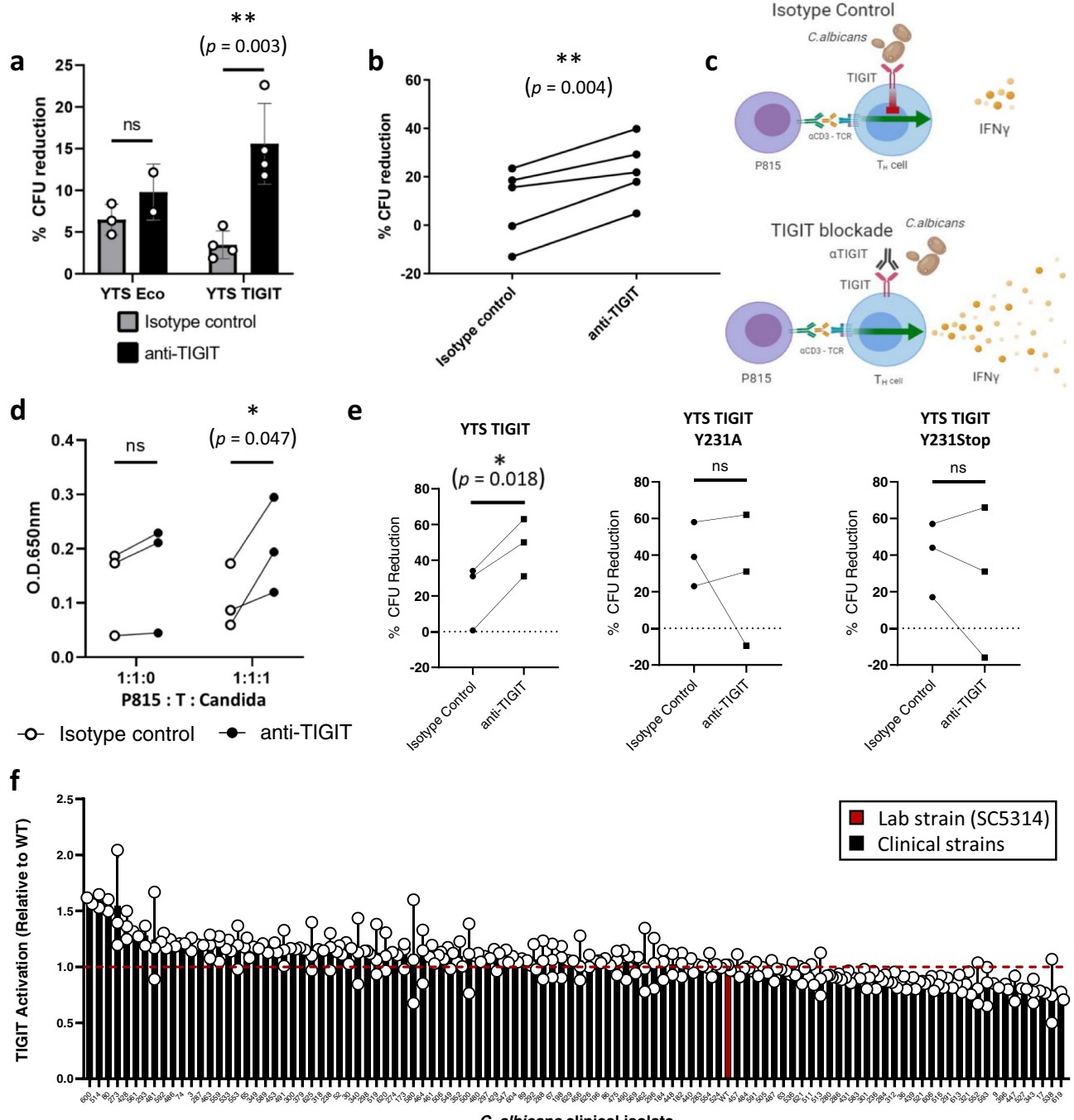

**Fig. 2 *C. albicans* binding of TIGIT is functional and leads to inhibition of NK and T cells. a** Cytotoxicity assay of *C. albicans* yeast cells using the YTS NK cell line either expressing TIGIT (YTS TIGIT) or not expressing it (YTS Eco). The YTS cells were either blocked using anti-TIGIT antibodies (black bars) or not blocked (gray bars). $n = 2$–4 independent experiments. Data are presented as mean values ± SD. **b** Cytotoxicity assay of *C. albicans* yeast cells using primary NK cells isolated from different human donors. The NK cells were blocked or not using anti-TIGIT antibodies. Each line represents an independent human donor. **c** A Diagram depicting the in-vitro T-cell activation model used in (**d**). **d** T cell activation by *C. albicans* model. CD4+TIGIT+ T cells were isolated from human donors. The T cells were activated using anti-CD3-antibody-coated P815 cells in the presence or absence of *C. albicans* cells and the presence (black dots) or absence (white dots) of a TIGIT-blocking antibody. A quantification of 3 independent experiments is presented. **e** *C. albicans* cytotoxicity assay using the YTS NK cell line expressing either WT TIGIT (YTS TIGIT) or a mutated version of TIGIT (YTS TIGIT Y231A or YTS TIGIT Y231Stop). The YTS cells were either blocked using anti-TIGIT antibodies or not. A quantification of 3 independent experiments is presented. **f** TIGIT activation was assayed using the murine thymoma cell line BW expressing a chimeric TIGIT-ζ-chain receptor. The BW cells were co-incubated for 48 h in the presence of the WT *C. albicans* lab strain SC5314 (red bar) or clinical *C. albicans* strains isolated from a cohort of human invasive candidiasis patients (black bars). TIGIT activation was measured using ELISA for quantification of IL-2 secreted by the activated BW cells. $n = 104$ biologically independent samples containing 2–3 technical repeats examined over 2 independent experiments. Data are presented as mean values ± SD. For (**a**, **b**, **d**, **e**)) significance was tested using Student's *t* test (two-tailed and unpaired for (**a**), two-tailed and paired for (**b**, **e**), and one-tailed and paired for (**d**)). For (**f**), one-way ANOVA with correction for multiple comparisons using Dunnett's test was used. ns = not-significant, * = $p < 0.05$, ** = $p < 0.01$.

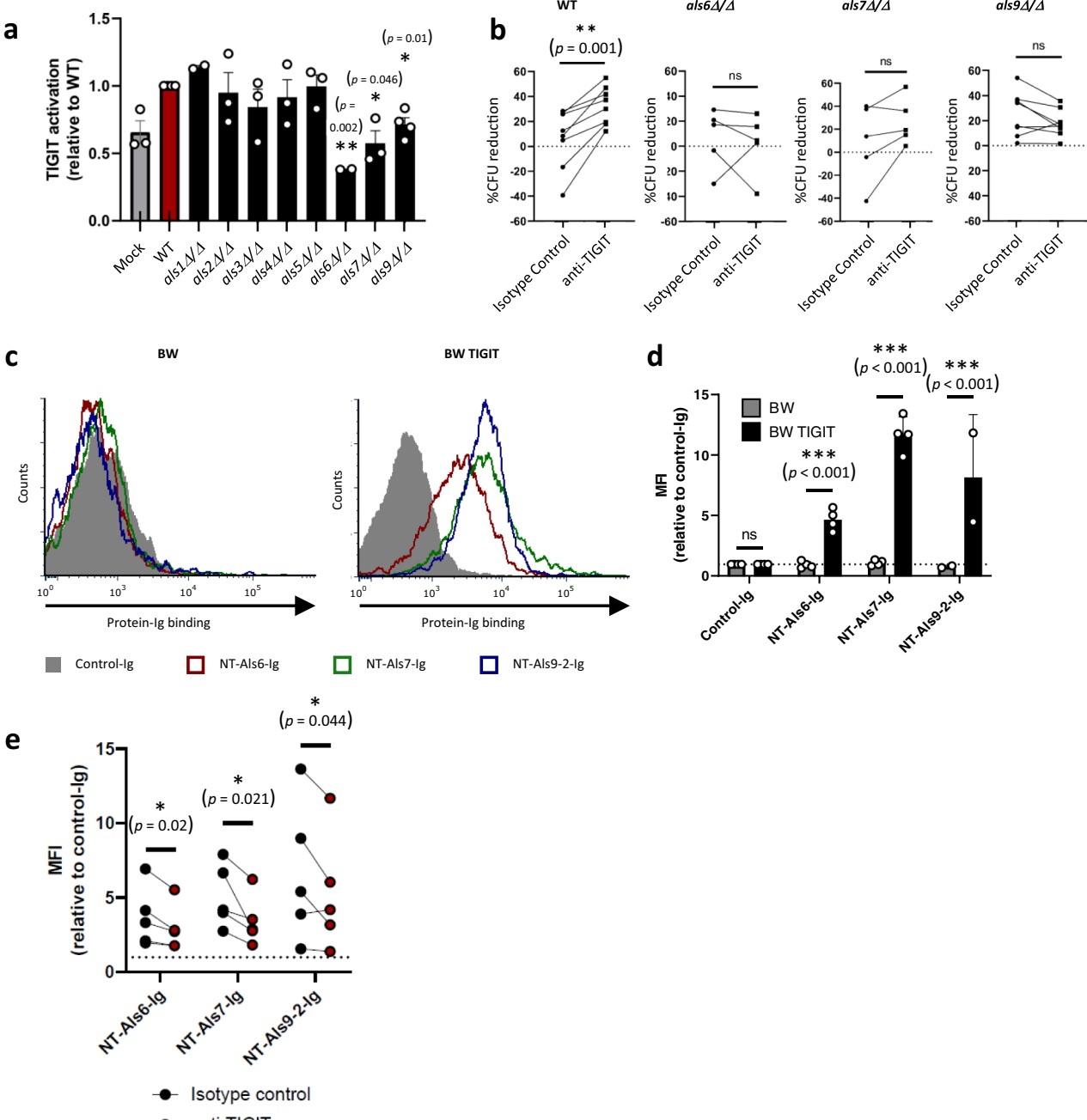

**Fig. 3 Als proteins of *C. albicans* are fungal TIGIT ligands. a** TIGIT activation was assayed using the murine thymoma cell line BW expressing a chimeric TIGIT-ζ-chain receptor. The BW cells were co-incubated for 48 h in the presence or absence of the WT *C. albicans* strain SC5314 or mutant strains deleted for members of the Als protein family. TIGIT activation was measured using ELISA for IL-2 secreted by the activated BW cells. $n = 2$–4 independent experiments. Values are shown relative to the TIGIT activation abilities of the WT strain. **b** Cytotoxicity assay of *C. albicans* strain SC5314 or mutant strains deleted for members of the Als protein family using primary NK cells isolated from different human donors. The NK cells were blocked or not using an anti-TIGIT antibody. Each line represents a different human donor. **c** Flow cytometry staining of BW and BW TIGIT cells using NT-Als6-Ig (red empty histogram), NT-Als7-Ig (green empty histogram), NT-Als9-2-Ig (blue empty histogram) or a negative control protein (filled gray histogram). One representative experiment out of 2–4 is presented. **d** Quantification of the results presented in (**c**). $n = 2$–4 independent experiments. Data are presented as mean values ± SEM. **e** Quantification of flow cytometry staining of BW TIGIT cells using either NT-Als6-Ig, NT-Als7-Ig, NT-Als9-2-Ig in the presence of an anti-TIGIT antibody (red dots) or an isotype control antibody (black dots). Each line represents an independent experiment. Error bars represent the standard error of the mean. Significance was tested using Student's *t* test (two-tailed unpaired for (**a**, **d**), two-tailed and paired for (**b**), and one-tailed and paired for (**e**)). ns = not-significant, * = $p < 0.05$, ** = $p < 0.01$, *** = $p < 0.005$.

strains *als6Δ/Δ*, *als7Δ/Δ* or *als9Δ/Δ*, were not (Fig. 3b). As deletions of these Als proteins reduced the inhibitory effect of TIGIT activation to insignificant levels, we conclude that they are TIGIT ligands expressed on *C. albicans*.

To test whether Als6, Als7 and Als9 directly interact with TIGIT we prepared fusion proteins of all of them fused to the Fc domain of human IgG$_1$. Due to the size of Als proteins and their potential for aggregation we used only their N′-terminal (NT)

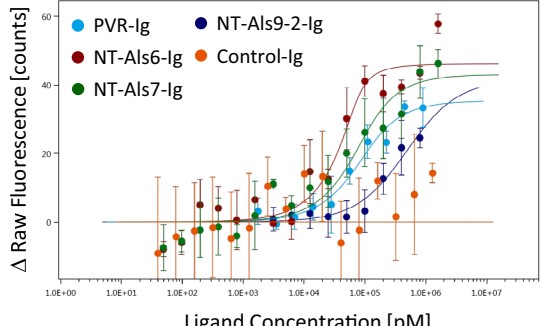

| Receptor | Ligand | $K_D$ [nM] |
|---|---|---|
| TIGIT-Ig | PVR-Ig | 49±22 |
| TIGIT-Ig | NT-Als6-Ig | 6±8 |
| TIGIT-Ig | NT-Als7-Ig | 42±27 |
| TIGIT-Ig | NT-Als9-2-Ig | 455±299 |
| TIGIT-Ig | Control-Ig | NB |

**Fig. 4 Purified Als fusion proteins are direct TIGIT ligands.** A microscale Thermophoresis experiment analyzing the interactions between fluorescently labeled TIGIT-Ig and either PVR-Ig (teal), NT-Als6-Ig (red), NT-Als7-Ig (green), NT-Als9-2-Ig (dark blue) or a negative control protein fused to an Ig tag (Control-Ig, orange). Shown are a graph and a summary table. $n = 2$–4 replicates derived from two biologically independent experiments. Error bars represent the standard error of the mean. NB = No Binding.

domain. This domain was also chosen due to the fact that it was previously shown to be the main ligand-binding domain and sufficient for ligand binding[48–50]. We successfully generated Ig-fusion proteins of Als6, Als7, and Als9. As *C. albicans* codes for two functionally distinct alleles of Als9, Als9-1, and Als9-2[51], we generated Ig-fusion proteins of both.

We used flow cytometry to check whether the various Als-Ig-fusion proteins directly bind TIGIT by staining parental BW cells (which do not express TIGIT (Supplementary Fig. 1E)) and BW cells transfected to express TIGIT (Fig. S1F). NT-Als6-Ig, NT-Als7-Ig, and NT-Als9-2-Ig stained the BW TIGIT-expressing cells specifically, while NT-Als9-1-Ig did not stain any of the cells (Fig. 3c, quantified in Fig. 3d, and Supplementary Fig. 2). The differential binding of TIGIT by the two ALS9 alleles is similar to previous reports of distinct ligands and functions of the two gene copies[51].

To check whether the anti-TIGIT antibody we use directly blocks the interaction between TIGIT and the Als proteins we again stained BW TIGIT cells with either NT-Als6-Ig, NT-Als7-Ig, or NT-Als9-2-Ig, but in the presence of the TIGIT blocking antibody or an isotype control antibody. As expected, the presence of the anti-TIGIT antibody significantly reduced the binding of all examined NT-Als-Ig proteins to the TIGIT-expressing cells (Fig. 3e).

To validate these direct interactions without possible interference from other proteins on the surface of mammalian or fungal cells or the use of antibodies, and to determine their affinities, we used Microscale Thermophoresis (MST) and compared the TIGIT-Als binding to the binding of TIGIT and one of its human ligands, PVR[25–27]. A direct, denaturation-sensitive interaction between TIGIT and the three Als ligands: NT-Als6-Ig, NT-Als7-Ig, and NT-Als9-2-Ig was observed, while a negative control protein fused to an identical Ig tag did not show any significant binding (Fig. 4). These interactions were all within an order of magnitude of the TIGIT-PVR interaction, leading us to conclude that the observed interactions are most likely strong enough to be biologically relevant. Interestingly, the relative strength of these interactions was similar to the relative reduction in TIGIT activation observed for the deletion mutants used in Fig. 3a, with *als6Δ/Δ* having the strongest effect and *als9Δ/Δ* having the weakest one.

**TIGIT manipulation of NK and T cells by *C. albicans* is an immune-evasion mechanism in mice.** We next wanted to test whether *C. albicans* recognition by TIGIT is a functional immune-evasion mechanism in vivo, or whether it is a host-beneficial mechanism that can protect it from infection related

immunopathology. We began by validating the relevance of NK and T cells in mice during invasive candidiasis, as previous works have observed contradicting results regarding this issue[15,18]. C56BL/6 mice were depleted or not of NK or T cells and then injected intravenously with *C. albicans* yeast cells. Mouse weight loss over time (and up to 2 weeks post-infection) and kidney fungal burden 48 h post-infection were monitored as these are the main accepted measures of disease severity in this model[52]. Weight reduction of 20% or more was regarded as the ethical endpoint of the experiment. A significant increase in mortality and fungal burden were observed after depletion of both NK and T cells (Fig. 5a, b), leading us to the conclusion that not only T cells, but also NK cells confer an important protective effect during invasive candidiasis.

To test whether the mouse TIGIT orthologue can also bind *C. albicans*, we generated a mouse TIGIT-Ig-fusion protein as was described for its human counterpart. *C. albicans* yeast cells were stained with this protein and analyzed using flow cytometry. As was observed for the human orthologue, the murine TIGIT-Ig protein successfully stained the *C. albicans* cells (Fig. 5c, quantified in d). We next wanted to validate the functionality of this interaction and to check whether a commercial αTIGIT antibody that was previously shown to block the interaction between murine TIGIT and its endogenous ligands[53] is also able to affect the interaction with the fungal ligands. We isolated murine NK cells from the spleens of C57BL/6 mice and co-incubated them with *C. albicans* cells in the presence or absence of this antibody and measured the reduction in CFU relative to fungal cells grown in absence of NK cells. In all tested mice, TIGIT blockade led to an increase in the anti-fungal capacity of the mouse NK cells (Fig. 5e).

Following these results we concluded that the murine orthologue of TIGIT also binds *C. albicans*, that this interaction leads to NK cell inhibition, and that the αTIGIT antibody we used is able to block the interaction between murine TIGIT and its fungal ligands and could be used in in vivo mouse experiments.

**The Als-mediated immune evasion mechanism can be prevented using TIGIT-targeted immunotherapy in mice.** Finally, we validated in vivo that Als6, Als7, and Als9-2 are indeed TIGIT ligands by infecting mice intravenously with *als6Δ/Δ*, *als7Δ/Δ*, or *als9Δ/Δ* mutants or the WT strain of *C. albicans* and measuring both mouse survival over time and fungal burden within the kidneys 48 h post-infection. In order to examine whether immune checkpoint blockade with αTIGIT antibody has a therapeutic potential we also treated the mice with a TIGIT-blocking antibody or mock treatment. As expected by their role as

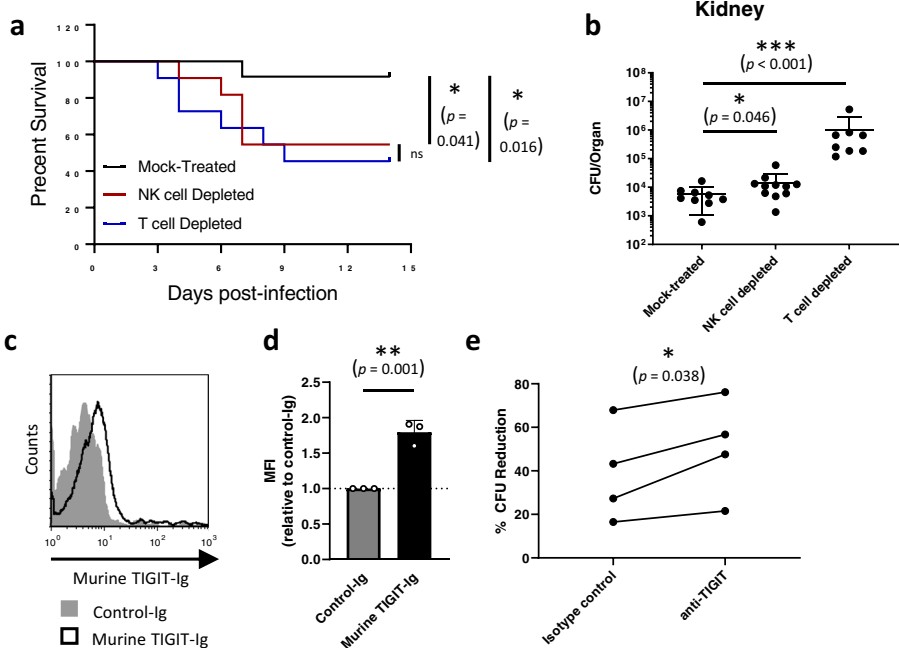

**Fig. 5 TIGIT manipulation of NK and T cells by *C. albicans* is an immune-evasion mechanism in mice. a** Survival of C57BL/6 mice infected intravenously with *C. albicans* cells and depleted for NK cells (red line), T cells (blue line) or mock-depleted (black line). Each line represents 10–12 mice from 2 independent experiments. **b** *C. albicans* burden in the kidney 48 h post I.V. infection of C57BL/6 mice. The mice were depleted for NK cells, T cells or mock-depleted. Kidneys were harvested, processed, and seeded on Sabouraud dextrose agar plates. *n* = 8–11 animals examined over 2 independent experiments. Data are presented as mean values ± SD. **c** Flow cytometry staining of *C. albicans* SC5314 yeast cells using Murine TIGIT-Ig (Black empty histogram) or a negative control protein (filled gray histogram). One representative experiment out of 3 is presented. **d** Quantification of the results presented in (**c**). *n* = 3 independent experiments. Data are presented as mean values ± SEM. **e** Cytotoxicity assay of *C. albicans* yeast cells using splenic NK cells isolated from mice. The NK cells were blocked or not using anti-TIGIT antibodies. Each line represents cells from one mouse collected during one of two independent experiments. For the survival assay significance was tested using a Mantel–Cox log-rank test. For the fungal burden significance was tested using a two-tailed Mann–Whitney test. For the flow cytometry and NK cytotoxicity assay a two-tailed Student's *t* test was used. ns = not-significant, * = *p* < 0.05, ** = *p* < 0.01, *** = *p* < 0.001.

pathogenicity factors, strains lacking Als6 and Als7 showed reduced pathogenicity, but under our experimental conditions *als9Δ/Δ* appeared to not be significantly less pathogenic relative to the WT strain (Fig. 6a). We next compared the effects of the different mutants in the presence of a TIGIT-blocking antibody. TIGIT blockade significantly reduced mortality and fungal burden during infection with the WT strain. Importantly, TIGIT blockade did not reduce fungal burden during infection when the Als mutant strains were used (Fig. 6b, c). This suggests that the effects of TIGIT blockade are a consequence of its inhibition of the Als-mediated immune evasion mechanism of *C. albicans*, and not a consequence of the general immune-disinhibitory effects of immune checkpoint inhibitors. Surprisingly, TIGIT blockade during *C. albicans als6Δ/Δ* infection even led to a paradoxical effect as it increased the fungal burden relative to the untreated mice. The reasons for that are yet unclear. In contrast, no difference was observed with and without anti-TIGIT antibody when *als7Δ/Δ* and *als9Δ/Δ* strains were used. While no significant mortality was observed during infection with the *als6Δ/Δ* and *als7Δ/Δ* strains, a significant kidney fungal burden was observed, indicative of successful infection of the mice. The *als9Δ/Δ* strain did lead to significant mortality and kidney fungal burden in the infected mice, and surprisingly did not show a significantly reduced pathogenicity relative to the WT strain. With that in mind, the lack of effect of TIGIT blockade during infection with it suggests that it does play a role in TIGIT-mediated immune evasion. The maintained virulence of this strain is probably since Als9 plays different roles at different infection sites and times. While Als proteins mostly function as adhesion molecules, it was shown that their deletion can also lead to reduced adhesion to

some surfaces[54]. In this case, the net effect Als9 deletion, which consists of its pro- and anti-adhesion effects and its TIGIT-activating abilities, is neutral.

As the deletion of Als family members completely abolished the effects of TIGIT blockade, we concluded that the effect of said blockade during infection with the WT strain is due to the interaction of TIGIT with Als proteins and not due to interference of the blocking antibody with the interactions between TIGIT and its endogenous ligands.

These observations led us to the conclusion that Als6, Als7, and Als9-2 are TIGIT ligands that act as immune evasion proteins in vivo, and that this interaction can be prevented via TIGIT blockade.

## Discussion

Fungal pathogens are responsible for over 1.5 million deaths annually[55] with the most prevalent fungal pathogen being *Candida albicans*, against which only limited treatment options exist. New treatments, together with a better understanding of the anti-fungal immune response, are therefore greatly needed. Specifically, the role of NK cells in the anti-fungal immune response is still unclear. While their importance during invasive candidiasis in mice has been demonstrated, whether they confer a protective or detrimental effect and the mechanisms behind these roles are still under debate[3,15,18].

Here we focus on recognition of *C. albicans* by NK and T cells and in particular on how *C. albicans* evades this immune recognition. We identified an immune-checkpoint-receptor-mediated immune evasion mechanism of fungi: an interaction between the immune checkpoint receptor TIGIT and *C. albicans*.

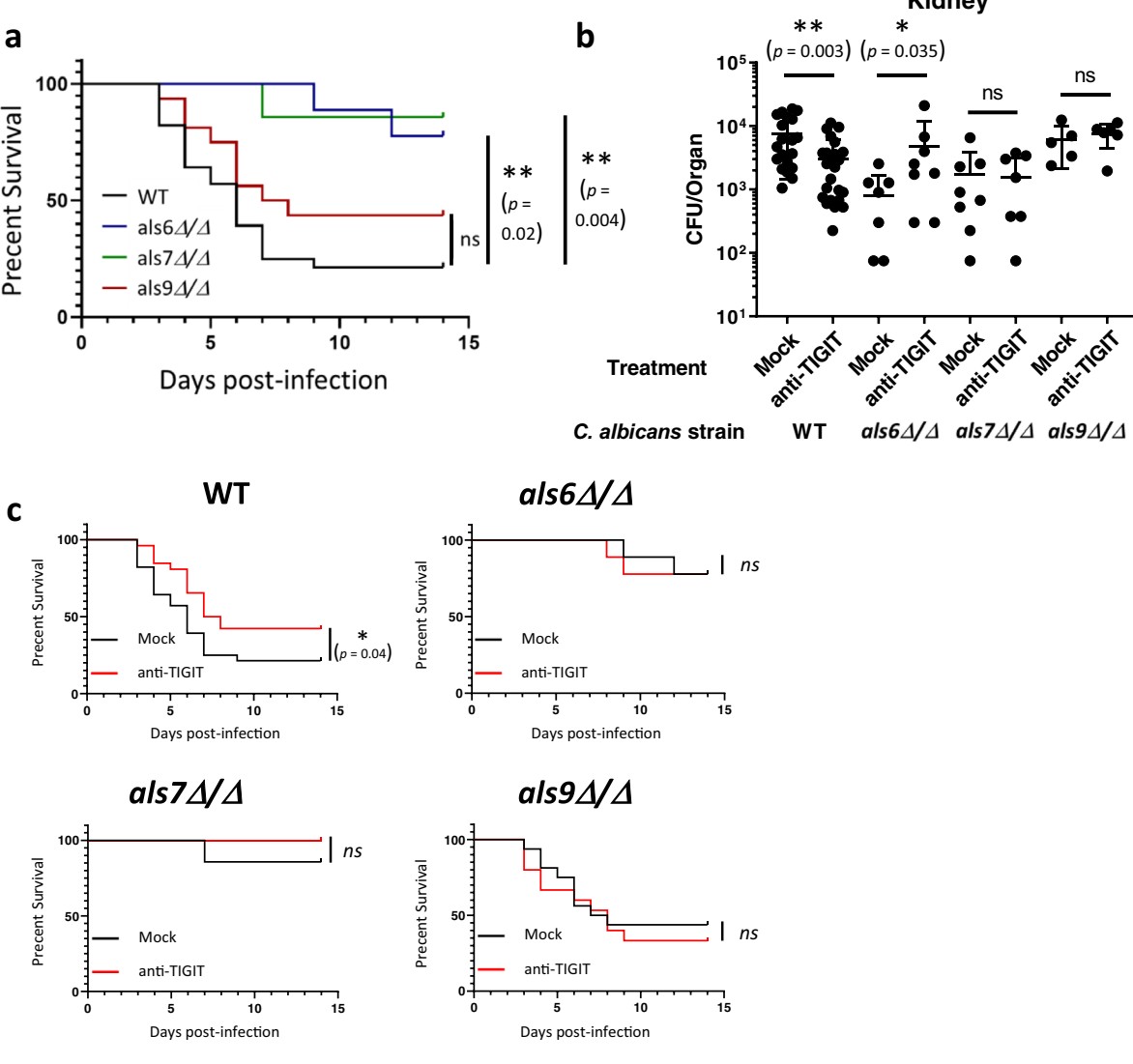

**Fig. 6 Als-mediated immune evasion can be prevented using TIGIT-targeted immunotherapy in mice. a** Survival of mice infected with $6.5 \times 10^5$ CFU/ mouse of the WT *C. albicans* strain SC5314 (black line) or a mutant strain deleted for ALS6 (blue line), ALS7 (green line), or ALS9 (red line). Each line represents 7–28 mice from 2–5 independent experiments. **b** *C. albicans* burden in the kidneys 48 h post I.V. infection of C57BL/6 mice. The mice were infected with either WT or ALS6, ALS7 or ALS9-deleted fungal cells and either treated or not with a TIGIT-blocking antibody. Kidneys were harvested, processed, and seeded on Sabouraud dextrose agar plates. *n* = 5–25 animals examined over 2–5 independent experiments. Data are presented as mean values ± SD. **c** Survival of mice infected with $6.5 \times 10^5$ CFU/mouse of the WT *C. albicans* strain SC5314 or a mutant strain deleted for ALS6, ALS7, or ALS9, and either mock treated (black line) or treated with a TIGIT-blocking antibody (red line). Each line represents 7–28 mice from 2–5 independent experiments. Data for the mock-treated mice is shared with (**a**). Error bars represent the standard error. For the fungal burden, significance was tested using a two-tailed Mann–Whitney test. For the survival assay significance was tested using Mantel–Cox log-rank test. ns = not-significant, * = *p* < 0.05, ** = *p* < 0.01.

We further show that this interaction leads to the inhibition of antifungal NK and T cell activities in both humans and mice, both in-vitro and in vivo. We identify the fungal ligands responsible for the manipulation of TIGIT as members of the Als adhesin proteins: Als6, Als7, and Als9-2. Most importantly, we show that a TIGIT-blocking antibody, similar to checkpoint inhibitors currently being used in clinical trials for the treatment of various cancers[37–39], can be successfully used for the treatment of *C. albicans* infection.

Another surprising finding rising from the data presented here is the paradoxical effect of TIGIT blockade during infection with the *als6Δ/Δ C. albicans* strain but not with other Als mutant strains, in which we observed increased fungal burden following TIGIT blocking. This surprising effect is yet unclear and merits additional study but suggests the existence of a complicated

mechanism behind Als-mediated TIGIT manipulation. Another interesting observation is the fact that even though three separate Als ligands were identified, each one able to directly bind TIGIT independently, deletion of just one is enough to completely abolish their effect on TIGIT. This observation points at a combined and dependent mechanism of action of the three ligands. Such dependence could rely on co-regulation within the Als family, or on physical interaction and the existence of a single TIGIT ligands complex containing several different Als proteins, as was previously suggested for other Als binding partners[48]. While our data show the role of three Als proteins as TIGIT ligands, it does not negate the possibility that other fungal proteins, Als or other, can also bind and perhaps activate TIGIT. Our initial experiment that identifies Als6, Als7 and Als9 can only identify non-redundant TIGIT-activating proteins. It is still

possible that additional proteins (including Als1-5) bind TIGIT and are involved in its activation in a redundant and Als6/7/9-dependent fashion (as deletion of these proteins completely abolished the effect of TIGIT blockade in vivo, hinting against additional redundant pathways for it).

Als proteins usually bind their targets using a specific motif in their N-terminal domain[48–50]. The Ig-fusion proteins we generated only contain this N-terminal domain and are sufficient for TIGIT binding. This suggests that the structural mechanism responsible for the aforementioned interaction is similar, but a final determination of this requires additional structural data.

Several cancers use TIGIT to impair NK and T cell responses either directly, or indirectly through the use of the tumor-associated bacterium *Fusobacterium nucleatum*[28,31,32,34].

Here we show that manipulation of immune checkpoint proteins for immune evasion is much more widespread, and that not only tumors, but also microbial pathogens can manipulate TIGIT to evade the immune response. Although this work focuses on the fungal pathogen *C. albicans*, it seems feasible that additional organisms will be found that also use TIGIT or similar immune checkpoint receptors to inhibit immune responses. The fact that the intracellular signaling cascade initiated by fungal TIGIT activation seems to be ITIM-based and similar to the cascade described for activation by endogenous TIGIT ligands, also hints that immune checkpoint manipulation might be simple to evolve, and overall an accessible evolutionary immune evasion mechanism. For example, as shown here, it is possible that *C. parapsilosis* also interacts with TIGIT. Indeed, this species was recently shown to also code for Als family proteins[56], which, as we show here, function as TIGIT ligands.

We also tested and showed that many *C. albicans* clinical isolates have the ability to activate TIGIT. This ability was present in all clinical strains but at different levels. This difference could originate from variance in Als expression levels, or from other more complicated regulatory effects and would be interesting to study in the future.

Recent works have highlighted the important role of *C. albicans* in shaping our immune response[4,5,57]. TIGIT is expressed on various immune cells under healthy steady-state conditions. Thus, it is likely that TIGIT activation by *C. albicans* takes place under these physiological conditions as well. Such activation could possibly affect local immune cells in the gut to generate an anti-inflammatory effect, or hinder the ability of the local immune cells to control *C. albicans* and so increase the basal levels of local inflammation due to increased *C. albicans* burden. The posible role of TIGIT-*C. albicans* interaction during healthy physiological conditions and its effects on the general anti-fungal immune response remains an open and interesting question. A specifically interesting site is the mucosal barrier, as *C.albicans* is a common pathogen of such barriers in humans in diseases such as vulvovaginal candidiasis or oral thrush. While we show that in systemic candidasis Als-mediated TIGIT manipulation is detrimental to the host, the case could be different in mucosal immunology. A relevant recent study found that an aberrant $T_H1$ response, which is commonly mediated by both NK and $T_H1$ cells, is in fact driving the pathologic proccess during mucosal candidiasis[58].

Increased *C. albicans* density in the gut also often accompanies inflammatory bowel diseases[57]. It will therefore be interesting to test TIGIT interactions with *C. albicans* in this family of diseases as well. Furthermore, since *C. albicans* is a common gastro-intestinal commensal organism, it is often also present in the vicinity of several cancers such as colorectal cancers or oral squamous cell carcinomas[59]. It is possible that, through TIGIT manipulation, *C. albicans* can affect the tumor microenviroment and confer some tumors with immune evasion capabilities,

similar to the effects elicited by *F. nucleatum* during colorectal cancer[34].

In conclusion, we show here a direct, immune checkpoint receptor-mediated, immune evasion mechanism of fungi. We demonstrate that *C. albicans* inhibits NK cells via manipulation of the immune checkpoint receptor TIGIT using adhesion proteins belonging to the Als family. We further show that this inhibition takes place in both humans and mice, and that immune checkpoint blocking antibodies aimed against TIGIT can be used for the treatment of *C. albicans* infection models. In the future, using immunotherapy for the treatment of infectious diseases could provide additional tools in the fight against antibiotic-resistant pathogens, adding a new trick to our bag in the host-pathogen arms race.

## Methods

**Mice**. Mice used in this study were male C57BL/6 mice aged 6–8 weeks acquired from Envigo Israel. The mice were test-naïve and were group-housed under specific pathogen-free (SPF) conditions prior to their use. Littermates were randomly allocated to the different experimental groups. All experiments were done in the SPF unit of the Hebrew University-Hadassah Medical School (Ein-Kerem, Jerusalem) in accordance with the guidelines of the Declaration of Helsinki and the local research ethics committee.

**Human patients**. All work involving human patients was approved by the Tel Aviv Sourasky Medical Center Institutional Ethics Committee (approval number 0729-16). The need for informed consent was waived given the retrospective observational nature of this study. No compensation was offered to the participants of this study.

**Primary cells and cell lines**. Primary NK and T cells were isolated from blood donations collected from healthy individuals and under the approval of the institutional Helsinki committee (Helsinki number 0030-12-HMO). Peripheral blood mononuclear cells (PBMCs) were produced from heparin-treated blood after centrifugation in the presence of Lymphoprep (STEMCELL Technologies). TIGIT-expressing CD4+ cells were isolated by plating PBMCs and identifying CD4+TIGIT+ clones using flow cytometry with 0.1 µg/50,000 cells of anti-human CD4-PE and anti-human TIGIT-APC antibodies or their relevant isotype controls (Rat IgG2b-PE and mouse IgG2a-APC) (all from BioLegend). NK cells were isolated from PBMCs using EasySep human NK cell enrichment kit (STEMCELL Technologies). The cells were co-cultured in U-bottomed 96-well plates with irradiated (6000 RAD) PBMCs from two independent donors ($5 \times 10^4$/well per donor) and irradiated (6000 RAD) RPMI-8866 cells ($5 \times 10^3$/well). The cells were grown in DMEM:F12 medium mix (70:30) with 10% human serum (Sigma-Aldrich), 1 mM sodium pyruvate (Biological Industries), 2 mM glutamine (Biological Industries), nonessential amino acids (Biological Industries), 100 U/ml penicillin (Biological Industries), 0.1 mg/ml streptomycin (Biological Industries), 500 U/ml rhIL-2 (PeproTech) and 20 mg/ml PHA (Sigma-Aldrich). The cells were grown at 37 °C and 5% $CO_2$. NK cell identity was validated using duel staining with 0.1 µg/50,000 cells of anti-human-CD56-PE (clone HCD56) and anti-human-CD3-FITC (clone HIT3a) antibodies or their relevant isotype controls (mouse IgG1-PE and mouse IgG2a-FITC) (all from BioLegend) and flow cytometry.

Murine splenic NK cells were produced by first extracting splenocytes from freshly harvested spleens of experiment-naïve heterozygous NCR1+/GFP C57BL/6 mice. The splenocytes were cleaned from contaminating cells using a 5 min incubation in ACK medium (37 mM $NH_4Cl$, $10 mM KHCO_3$, 0.1 M EDTA) at room temperature followed by a wash (515 G for 5 min at 4 °C). Splenic NK cells were isolated from the splenocytes using EasySep mouse NK cell enrichment kit (STEMCELL Technologies). Purity and TIGIT expression were validated using flow cytometry. NK cells were recognized by their GFP expression and TIGIT expression was observed by staining with an APC-conjugated anti-TIGIT antibody.

Cell lines used in this study were HEK293T cells, YTS Eco cells, YTS TIGIT cells, P815 cells, YTS TIGIT Y231A cells, YTS TIGIT Y231Stop cells, BW cells, BW TIGIT cells and RPMI-8866 cells. The generation of BW TIGIT, YTS TIGIT, YTS TIGIT Y231A, and YTS TIGIT Y231Stop cells was previously described[25]. The BW, YTS, HEK293T cells, P815 cells and RPMI-8866 cells were a kind gift from Prof. Jack L. Strominger. All cells were grown in RPMI-1640 media (Sigma-Aldrich) except for HEK293T cells which were grown in Dulbecco's modified Eagle's medium (DMEM, Sigma-Aldrich). All cell line cultures were supplemented with 10% inactivated fetal bovine serum (Sigma-Aldrich), 1 mM sodium pyruvate (Biological Industries), 2 mM glutamine (Biological Industries), nonessential amino acids (Biological Industries), 100 U/ml penicillin (Biological Industries) and 0.1 mg/ml streptomycin (Biological Industries). Unless noted otherwise, all experiments including mammalian cells were conducted in these media.

**Microbial strains**. The fungal species and strains used in this study were *Candida albicans* SC5314, *Candida glabrata* BG2, *Candida parapsilosis*, *Candida krusei*, or clinical isolates of *Cryptococcus neoformans* and *Cryptococcus gattii*. *C. albicans* deletion mutants used were *C. albicans* als1Δ/Δ1467, als2Δ/Δ 2757, als3Δ/Δ 1843, als4Δ/Δ 2034, als5Δ/Δ 2373, als6Δ/Δ 1420, als7Δ/Δ 1429 and als9Δ/Δ 2028, which are all derivatives of SC5314.

Unless written otherwise, all Candida species and strains were kept in −80 °C frozen glycerol stocks and grown regularly on Sabouraud dextrose agar plates (Sigma-Aldrich) for up to 4 weeks. Prior to an experiment the fungi were inoculated into Sabouraud dextrose broth (Sigma-Aldrich) and grown overnight at 30 °C under shaking and aerobic conditions. The overnight culture was diluted into fresh Sabouraud dextrose broth (1:50) and grown for an additional 2–4 h before its introduction into the experiment.

**Flow cytometry**. Mammalian or fungal cells were grown as described above. At the start of the experiment, the cells were washed three times in ice-cold 1xPBS. For each wash the conditions were 515 G (for mammalian cells) or 3000 G (for fungal cells) for 5 min at 4 °C. Following washes, the cells were counted using a hemocytometer and divided into U-bottomed 96-well plates to a concentration of 5 or $10 \times 10^4$ cells/well. When blocking antibodies were used, the cells were incubated in the presence of 2.5 µg/well of the relevant antibody for 30 min on ice. For stainings, each well was incubated in the presence of primary antibodies (0.25 µg/well) or Ig-fusion proteins (0.5–5 µg/well) diluted in FACS medium (1x PBS, 0.05% Bovine Serum Albumin, 0.05% NaN₃) for 1 h on ice. The negative control protein used for the experiments was either NKp46-D1-Ig, NKp46-D2-Ig, or NKG2D-Ig. In instances when the primary antibodies were not fluorophore-conjugated the cells were next washed one time with FACS medium, and then stained with 2nd antibodies (0.75 µg/well) for 30–45 min on ice. The secondary antibodies used were either Anti-human-Alexa-Fluor647, Anti-human-APC, or Anti-mouse-AlexaFluor647 (all from Jackson ImmunoResearch). Finally, the cells were washed 2 times in ice-cold FACS medium and analyzed using either a FACSCalibur machine (BD Biosciences) or a CytoFlex machine (Beckman-Coulter Life Sciences) and the FCS Express software (De Novo Software). Gating was performed as presented in Supplementary Fig. 3).

**Ig-fusion protein generation**. The extracellular portion of the fusion proteins used was cloned into a mammalian expression vector (pIRESpuro3) containing a mutated Fc domain of human IgG₁ adjacent to its integration site. The generation of human TIGIT-Ig, mouse TIGIT-Ig, NKp46-D1-Ig, NKp44-Ig, and PVR-Ig was described previously[25,60–62]. NT-Als6-Ig, NT-Als7-Ig, NT-Als9-1-Ig, and NT-Als9-2-Ig were generated by adding the signal peptide of human CD5 to the sequence of the relevant domains of the original proteins, flanking them with EcoRI and BamHI restriction sites and codon-optimizing the sequence for expression in human cells (full sequence available as Supplementary Data 1). The sequences were generated synthetically as gBlocks gene fragments (IDT). The expression vector was amplified in chemically competent DH5α *Escherichia coli* bacteria grown in Luria Broth at 37 °C and extracted from the bacteria using AccuPrep Plasmid Mini Extraction Kit (Bioneer Corporation). The vector was then transfected into HEK293T cells using the reagent TransIT-LT1 (Mirus Bio). Following a 48-h recovery the cells underwent selection using Puromycin (5 µg/ml). Surviving colonies were grown separately and measured for fusion protein secretion using ELISA assay performed on their growth media using an anti-human-IgG antibody (Jackson Immunoresearch, polyclonal). The clones secreting the highest amount of protein were propagated and eventually transferred into Low Protein BSA-Free medium (LPM, Biological Industries) complemented with 1 mM sodium pyruvate (Biological Industries), 2 mM glutamine (Biological Industries), nonessential amino acids (Biological Industries), 100 U/ml penicillin (Biological Industries) and 0.1 mg/ml streptomycin (Biological Industries. The medium was collected and Ig-fusion proteins were purified using a HiTrap Protein G HP column (Sigma-Aldrich) in a BioCAD High Pressure Perfusion Chromatography Station (Per-Septive Biosystems). The resulting proteins were buffer exchanged using dialysis bags into 1xPBS. Protein quality and purity were examined using sodium dodecyl sulfate–polyacrylamide gel electrophoresis (SDS/PAGE) followed by Coomassie staining of the gel using Imperial™ Protein Stain (ThermoFisher Scientific). Protein quantity was measured using a Pierce™ BCA Protein Assay Kit (ThermoFisher Scientific).

**ELISA**. All ELISA experiments were performed in high-binding clear F-bottomed 96-well ELISA plates (De-Groot group). For staining *C. albicans* cells using Ig-fusion proteins $5 \times 10^5$ cells were plated in each well, centrifuged (3000 G, 5 min, 4 °C) and left to bind to the well for 2 h in a stationary 30 °C incubator. The cells were then blocked using PBS-BSA (1% w/v bovine serum albumin diluted in 1xPBS) for 2 h at room temperature and washed 4 times using PBST (1xPBS supplemented with 0.05% Tween 20). Next the Ig-fusion proteins were diluted in PBS-BSA and added to the wells (2.5 µg/well, final volume 100 µl/well) and the plates were incubated for 2 h on ice. The negative control protein used for the experiments was NKp44-Ig. The plates were then washed 4 times with PBST and a detection antibody (Biotin-SP-AffiniPure Rabbit Anti-Human IgG, Jackson ImmunoResearch) diluted in PBS-BSA (1:7500) was added for 1 h at room temperature. Next the plates were washed 6 times with PBST, incubated for 30 min with Streptavidin-HRP (Jackson ImmunoResearch)

and again washed 6 times using PBST. Finally, the plates were developed using 3,3′,5,5′-tetramethylbenzidine (TMB) substrate (SouthernBiotech) and read at 650 nm.

For the deletion library screen a similar method was used but the fungal cells were grown directly in the ELISA plate wells overnight in Sabouraud dextrose broth (Sigma-Aldrich) and in a 30 °C stationary incubator. Following the above-mentioned protocol and in order to compensate for different growth rates or adherence strengths of the various mutants, the ELISA plates were washed 6 times with PBST and then 100 µl XTT/Menandione mix (0.5 gr/L and 1 µM, respectively, Sigma-Aldrich) was added to each well. The plates were covered in aluminum foil and incubated for 2 h in a stationary 37 °C incubator. Next, the plates were centrifuged (3000 G, 4 °C, 5 min), and 80 µl/well were transferred to a new plate and then read at 490 nm. For each well, the 650 nm read was normalized to the 490 nm read.

For the IL-2 ELISA following the BW assay or IFNγ ELISA following the T cell activation assay, a protocol similar to the one described above for Ig-fusion proteins was used, with the following changes: The ELISA plates were initially coated with 0.2 µg/well of 1xPBS-diluted anti-mouse-IL2 or anti-human IFNγ antibodies (Both from BioLegend) for 2 h in 37 °C or overnight in 4 °C. Instead of *C. albicans* cells, medium from the experimental wells was centrifuged (3000 G, 5 min, 4 °C) and the supernatant was added to the plates, and the detection antibody was biotinylated anti-mouse-IL2 or biotinylated anti-human IFNγ (both from BioLegend) from different clones than the coating antibody and a final concentration of 0.1 µg/well.

**Immunofluorescence microscopy**. *C. albicans* yeast cells were grown as described above. Hyphal cells were grown overnight at 37 °C with RPMI-1640 media (Sigma-Aldrich) supplemented with 10% inactivated fetal bovine serum (Sigma-Aldrich), 1 mM sodium pyruvate (Biological Industries), 2 mM glutamine (Biological Industries), nonessential amino acids (Biological Industries), 100 U/ml penicillin (Biological Industries) and 0.1 mg/ml streptomycin (Biological Industries). Each staining contained $1.5 \times 10^5$ cells. The cells were washed 3 times in ice-cold 1xPBS (3000 G, 4 °C, 5 min) and blocked using CAS-Block (ThermoFisher Scientific) for 1.5 h at 4 °C. The cells were washed again in 1xPBS and 15ug of Ig-fusion proteins diluted in CAS-Block were added. The negative control protein used for the experiments was NKp46-D1-Ig. The cells were incubated with the fusion proteins for 2 h at 4 °C and then washed twice in ice-cold 1xPBS. A 2nd antibody (APC-anti-human IgG, Jackson ImmunoResearch) was diluted 1:500 in 1xPBS and added for 1 h at 4 °C and then the cells were washed twice in ice-cold 1xPBS. The cells were incubated in the presence of FITC (0.1 mg/ml, diluted in 1xPBS) for 25 min, washed twice in ice-cold 1xPBS and fixated using 4% PFA (Bar-Naor Ltd) overnight. Finally, the cells were mounted on an 8-chamber slide (Bar-Naor) and visualized using an Olympus Fluoview FV1000 confocal microscope.

**Cytotoxicity assay**. Fungal and mammalian cells were grown as described above. The cells were washed 3 times in either sterile 1x PBS or RPMI-1640 and in the following conditions: 515 G (for mammalian cells) or 3000 G (for fungal cells) for 5 min and at 4 °C. Following the washes the cells were counted using a hemocytometer. Mammalian cells were then incubated in the presence of isotype control or TIGIT-blocking antibodies (1 µg/10x⁵ cells, diluted in RPMI-1640-based growth media described above) for 1 h on ice. Following the blocking stage, the effector mammalian cells were mixed with the target fungal cells in U-shaped 96-well plates ($5 \times 10^4$ mammalian cells, 1000 fungal cells) and in a final volume of 200 µl RPMI-1640-growth media (described above). The cells were co-incubated for 12–14 h in a stationary 37 °C, 5% CO₂ incubator and then serially diluted in 1xPBS and plated on Sabouraud dextrose agar plates (Sigma-Aldrich). The plates were incubated in a stationary 30 °C incubator and the number of colonies in each plate was measured after 24–48 h. The % Colony Forming Units (CFU) reduction was calculated by comparing the CFU counted in each Sabouraud dextrose agar plate to the CFU measured in a control plate in which a culture identical to all other samples but lacking effector mammalian cells was plated.

**T cell activation assay**. Primary T cells, P815 cells and *C. albicans* cells were isolated and cultured as described above. P815 cells were irradiated (6000 RAD), washed in RPMI media and incubated for 1 h on ice with isotype control (Mouse IgG2a) or anti-human CD3 antibodies (both from BioLegend). Antibody concentration was 0.1 µg/2.5 × 10³ cells. In parallel, the T cells were washed in RPMI media and incubated for 1 h on ice in the presence of either isotype control (Mouse IgG1) or anti-human TIGIT antibodies (BioLegend and in-house, respectively). Antibody concentration was 1 µg/2.5 × 10³ cells. Following the incubation both cell types were mixed in a U-bottomed 96-well plate format (2.5 × 10³ cells of each type per well). *C. albicans* cells were washed in sterile 1xPBS 3 times (3000 G, 4 °C, 5 min) and added to the relevant wells (2.5 × 10³ cells/well). The plates were incubated for 12 h in a 37 °C, 5% CO₂ stationary incubator and then frozen in −20 °C. At the day of ELISA, the plates were thawed, centrifuged (3000 G, 4 °C, 5 min), and the top media was taken for ELISA analysis using anti-human IFNγ antibodies (BioLegend) as described above.

**Murine invasive candidiasis model.** Male C57BL/6 mice, aged 6–8 weeks, and *C. albicans* cells, were grown as described above. On the day of experiment the fungal cells were washed 3 times in ice-cold 1xPBS and taken to the animal facility on ice. The mice were then injected with either $5 \times 10^5$ CFU/100 µl 1xPBS/mouse (for experiments measuring fungal burden or survival after NK cell or T cell depletion) or $6.5 \times 10^5$ CFU/100 µl PBS/mouse (for survival experiments without depletion). Antibodies used for TIGIT blockade (anti-mouse TIGIT clone 1G9, BioXCell, 100 µg/mouse), NK cell depletion (anti-mouse NK1.1 PK136, BioXCell, 25 µg/mouse) or T cell depletion (anti-mouse CD3 17A2, BioXCell, 100 µg/mouse) were diluted in 1xPBS for a final volume of 200 µl/mouse and injected intraperitoneally. Injection times were once every 48 h starting one day prior to infection.

For fungal burden experiments, the mice were sacrificed 48 h post-infection and relevant organs were harvested. The organs were physically homogenized, filtered through a 70 µm strainer, serially diluted in ice-cold 1xPBS and plated on Sabouraud dextrose agar plates (Sigma-Aldrich). The plates were incubated for 48 h in a stationary 30 °C incubator and once colonies were visible they were counted.

For survival experiments, the mice were weighted and clinically evaluated daily and euthanized once their weight was reduced to <80% of their starting weight. Mice that developed unusual clinical symptoms were noted, euthanized, and removed from the study

**BW assay.** BW and *C. albicans* cells were grown as described above. The cells were then washed three times in either sterile 1x PBS or RPMI-1640 using the following conditions: 515 G (mammalian cells)/3000 G (fungal cells), 5 min, 4 °C. The washed cells were then counted using a hemocytometer. The fungal cells were incubated in the presence of 50 µg/ml of Fluconazol (BioAvenir) for 1 h on ice and then the fungal and BW cells were mixed in F-bottomed 96-well plates in a final volume of 200 µl RPMI-1640 growth media described above and supplemented with 50 µg/ml of Fluconazol (BioAvenir). Each experimental well contained $5 \times 10^4$ BW TIGIT cells and $2.5 \times 10^4$ fungal cells, and had an identical duplicate well containing BW parental cells that do not express TIGIT. The cells were co-incubated for 48 h in a stationary 37 °C 5% $CO_2$ incubator. Finally, the experiment plates were frozen at −20 °C until the time of IL-2 secretion measurement. At that time the plates were thawed, centrifuged (3000 G, 5 min, 4 °C) and the top media underwent ELISA for IL-2 measurement as described above. The measured IL-2 levels for the BW TIGIT cells were compared to the relevant identical BW parental cells and normalized.

**Microscale thermophoresis (MST).** The microscale thermophoresis experiments were performed on a blue/red Monolith NT.115 machine (NanoTemper) using TIGIT-Ig protein labeled using the Monolith NT$^{TM}$ Protein Labeling Kit RED-NHS (NanoTemper). Capillary scan at 95% LED power was used to determine minimal fluorescence levels. All experiments were performed under 20% MST power for 30 s with 95% LED power in standard capillaries at 25 °C. Interaction between labeled TIGIT-Ig and non-labeled NT-Als6-Ig, NT-Als7-Ig, NT-Als9-Ig, PVR-Ig or a Control-Ig (NKp46-D1-Ig) was measured by maintaining constant levels of TIGIT-Ig and mixing it with a series of target proteins diluted in 1x PBS in serially decreasing concentrations. Binding was observed by measuring the change in fluorescence in each capillary containing a receptor-ligand mix. All identified interactions were validated to be denaturation-sensitive by adding DTT and SDS to the samples (final concentration 20 mM and 2% respectively), heating them (95 °C, 5 min), and reading them in the machine.

*Quantification and statistical analysis.* Statistical analysis was performed using either Prism 9 (GraphPad) or Excel (Microsoft). All of the relevant statistical data for the experiments including the statistical test used, value of n, definition of significance, etc. can be found in the figure legends or the relevant methods section.

**Reporting summary.** Further information on research design is available in the Nature Research Reporting Summary linked to this article.

## Data availability
All data generated in this study are provided in the Supplementary Information and Source Data files.

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

## Acknowledgements

The authors would like to thank Prof. Brendan Cormack for supplying critical fungal strains. We would like to also thank the Fungal Genomic Stock Center (FGSC) for providing the *C. albicans* deletion library[63]. Y.C.A. is supported by the Azrieli Foundation. This work was supported by the following grants awarded to O.M.: the Israel Innovation Authority Kamin grant 62615, the German-Israeli Foundation for Scientific Research and Development grant 1412-414.13/2017, the ICRF professorship grant, the ISF Israel-China grant 2554/18, the MOST-DKFZ grant 3-14931, and the Ministry of Science and Technology grant 3-14764 and by the Israel Science Foundation Moked grant 442-18 awarded to O.M., F.L.S., N.O., G.B. and R.B.A.

## Author contributions

Conceptualization, O.M. and Y.C.A.; Methodology, O.M., Y.C.A., B.I., A.D.C., and R.B.A.; Investigation, Y.C.A., T.L., B.I., A.D.C., A.E., and R.B.A.; Resources, A.E., L.L.H., M.K., and R.B.A.; Writing—Original Draft, O.M. and Y.C.A.; Writing—Review & Editing, Y.C.A., B.I., A.D.C., F.L.S., N.O., G.B., L.L.H., M.K., R.B.A., and O.M.; Visualization, Y.C.A. and R.B.A.; Supervision, O.M., F.L.S., N.O., G.B., and R.B.A.; Project Administration, O.M.; Funding Acquisition, O.M. and Y.C.A.

## Competing interests

The authors declare no competing interests.
