## [Peer Review File · Nature Communications]

Candida albicans evades NK cell elimination via binding of Agglutinin-Like Sequence proteins to the checkpoint receptor TIGITREVIEWER COMMENTS

Reviewer #1 (Remarks to the Author):

Charpak-Amikam et al report on the identification of the TIGIT receptor as the NK cell receptor binding to *Candida albicans* yeast and hyphae. Via this receptor, NK cells recognize the *C. albicans* cell wall-bound adhesion proteins Als6, Als7 and Als9. The ligands binding to this receptor lead to inhibition NK cell function, and in vivo studies in mice infected with *C. albicans* demonstrate higher mortality. As fungal infections are still a major cause of morbidity and mortality in different patient populations, the present data are of high interest for a better understanding of the host-fungal interaction and may provide the rationale for new therapeutic strategies.

Although the data are well presented and quite convincing, I would like to make several comments. The authors state that *C. albicans* binds NK cell TIGIT and inhibits NK cell activity in vitro, as demonstrated in Figure 2b. This was evaluated using 1) YTS cells, which naturally do not express TIGIT (YTS Eco), and in which blocking TIGIT does not have any impact 2) YTS-TIGIT cells in which the authors demonstrate that *C. albicans* proteins bind to the TIGIT receptor, resulting in the inhibition of the cellular antifungal activity. In contrast to YTS eco, the addition of α -TIGIT antibody increased (restored) the antifungal activity of YTS-TIGIT cells. Nevertheless, it remains unclear why the anti-*Candida* activity is lower in YTS Eco cells not expressing TIGIT compared to that of YTS-TIGIT cells after addition of an α -TIGIT antibody. Does the α -TIGIT antibody exhibit activating properties? Does the TIGIT receptor have activating properties and does the binding of the α -TIGIT antibody boost the antifungal activity of the transduced (YTS-TIGIT) cells? From the data presented in Figure 2a one cannot conclude that TIGIT is "functional and inhibitory for NK cells" (page 7, line 151). Authors, please address these comments/questions.

The y-axes in Figure 3b show different ranges, which means that the antifungal activity of the different mutants is different. Authors, please adjust the y-axes to allow a better comparison of the mutants and explain these differences.

The authors report that NK cells express TIGIT, which binds Als6/7/9 of *C. albicans* as ligands. The binding of these proteins (Als6/7/9) leads to the inhibition of NK cell activity, both in vitro and in vivo. α -TIGIT antibody counteracts this binding and therefore also NK cell inhibition by Als6/7/9, resulting in increased antifungal activity of NK cells. Mutant *C. albicans* strains lacking als 6, 7, or 9 (als6/7/9 Δ/Δ) do not inhibit NK cell activity, which demonstrates the importance of these fungal proteins in decreasing NK cell activity. In Figure 6c, it remains unclear why als6 Δ/Δ *C. albicans* and als7 Δ/Δ *C. albicans* (still expressing Als7 and 9 or Als6 and 9, respectively) do not inhibit NK cells, whereas als9 Δ/Δ (still expressing Als6 and 7) exhibits NK cell inhibition in vivo (Figure 6c), resulting in high mortality in *C. albicans* infected mice. Notably, the mortality is not significantly different from that of the wild type *C. albicans*. In addition, what is the impact on antifungal activity and mortality if combining different mutants (e.g., als6 Δ/Δ and als7 Δ/Δ or als6 Δ/Δ and als9 Δ/Δ etc)?

The authors state in the discussion section (page 14, line 297-298) that "deletion of just one of them [Als6, 7 or 9] is enough to completely abolish their effect on TIGIT" (page 14, line 297-298). However, data from Figure 6b and 6c show that deletion of als9 does not protect the mice from *C. albicans* infection, since the survival curve of the mice is similar of that of WT *C. albicans*, and therefore, this finding does not support this statement.

Reviewer #2 (Remarks to the Author):

The manuscript by Yoav Charpak-Amikam and colleagues aims to describe a new fungal immune evasion step of the human pathogenic yeast *Candida albicans*.

The authors approach how *Candida albicans* can inactivate and dampen the immune attack of human natural killer cells. The group focuses on the immune inhibitory receptor expressed on the surface of human NK cells, TIGIT, and they use a screening approach to identify the fungal ligands. Upon screening of a fungal deletion library, the authors identify *Candida* Als family proteins as ligands for

both human and mouse TIGIT receptors. In addition, they used an in vivo infection model where they approach the role of several Als isoforms, and furthermore they use a monoclonal antibody to block Als protein function.

This concept is of interest and may ultimately show a new avenue of fungal and in particular candida immune evasion. However, in the present form, additional work is required to clearly and unambiguously confirm the concept presented by the authors. Numerous additional controls experiments are necessary and need to be presented in order to provide a clear evidence that candida Als proteins interact with the immune inhibitory receptor TIGIT expressed on human and mouse NK, as well as T cells. In addition further proof is necessary to conclude that the TIGIT -Als interaction is the central avenue of fungal immune escape.

Issues in details:

Als has several family members. Does each Als proteins bind to the TIGIT receptors with similar affinity and is this interaction specific for human NK cells. What is the common binding motive.

Do all members of the ALS family proteins binds to TIGIT.

Also T cells express TIGIT receptors, what is the effect of the candida protein's on these central type of immune human cells.

Does the mAB used here block binding of the recombinant TIGIT to candida, also block the binding of each Als protein to TIGIT in the same manner. Does the antibody show the same inhibitory effect for all Als proteins.

The authors use fusion proteins in most experiments, they should include additional assays in which native untagged proteins are used, e.g. for immunoprecipitation.

Is this a uniform mechanism of fungal Als function? Do other fungi, like *Aspergillus fumigatus* use the same type of immune escape. Do *Aspergillus fumigatus* Als proteins, bind to the same human and mouse TIGIT ligands and do they have the same biological effects.

What is the signaling pathway initiated by Als proteins and what is the cell response. Is the effect the same in NK cells and in T lymphocytes? The immune dampening effect need to be evaluated in greater detail.

Also in this regard the Als-TIGIT Interaction may be relevant for mediated NK cell activity. But the effect on T cells needs to be included. In this regard, if the concept of Als-TIGIT interaction is really important than the effect on the host immune response mediated by NK and by T cells needs to be explored

Figure 1. One problem in this figure is presented in panels 1b, 1c, and 1d by the presentation of relative values. What are the real binding data. Like in 1c, a value two times about control may not be a dramatic high affinity binding.

For panels 1e and 1f. Whether controls include like using the mAB to block the interaction. How is the effect on binding to the various als knock out mutants. In particular given the redundancy of the various Als proteins this aspect needs to be addressed .

Figure 2. What is the meaning of the clinical strains. The clinical strains are an impressive selection. It would be nice to compare the expression levels of the various Als isoforms at least in several of the high and low effective isolates.

Figure 3. The effect TIGIT activation her all nine als knock out mutant were assessed What is the contribution of each als isoform on expression level and on TIGIT mediated activation. Please show also the raw data and not only the relative effects.

Figure 4. The effect a control protein (PVR-Ig) is comparable to Als7 protein. Does this reflect a specific interaction and high affinity binding? As negative control and lacking IgG domain needs to be included. In addition, k_{on} and k_{off} values and rates need to presented. A critical interpretation of this results is that the also Ig fusion proteins show related binding at the human ligand PVR Ig. However, a negative control needs to be presented.

Figure 5. The effect Figure 5a and 5b:

Figure 5d again please present the experimental data without the calculiton of the value relative to control. What is the effect in T cell depleted animals.

Figure 6.

The effect of the various *C. albicans* alsI knock out mutant in the in vivo infection model need to be evaluated and discussed first. Then the role of mAB can be addressed first.

What is the reason on selecting got the three mutants, als6, als7 and als 9. Additional control like revertant strains for each knock out mutant studied need to be presented.

Language: The text needs revision in language and also in content. Some of the statements sound rather dramatic and in the end are not absolutely right.

E.g. The point 'that *Candida albicans* is ... one of the deadliest causes of bloodstream infections' sounds very dramatic but is not complement right (page 2, abstract first sentence). Do the authors really want to say that other pathogenic bacteria like *Streptococcus pneumoniae*, *S pyogenes*, *Staphylococcus aureus* are not relevant. In particular not to mention the current COVID-19 situation which gives another example on severay of blood stream infections.

Reviewer #1 (Remarks to the Author):

Charpak-Amikam et al report on the identification of the TIGIT receptor as the NK cell receptor binding to *Candida albicans* yeast and hyphae. Via this receptor, NK cells recognize the *C. albicans* cell wall-bound adhesion proteins Als6, Als7 and Als9. The ligands binding to this receptor lead to inhibition NK cell function, and in vivo studies in mice infected with *C. albicans* demonstrate higher mortality. As fungal infections are still a major cause of morbidity and mortality in different patient populations, the present data are of high interest for a better understanding of the host-fungal interaction and may provide the rationale for new therapeutic strategies.

Although the data are well presented and quite convincing, I would like to make several comments.

We thank reviewer #1 for the thoughtful and interesting review. Following reviewer 1 and 2 comments 6 new figures were added (fig. 2c,d,e, 3e and S1c-d) and 5 more were modified and improved (fig. 1d, 3b, 4 and 5a-b). We have amended the manuscript in accordance with the following comments:

1. The authors state that *C. albicans* binds NK cell TIGIT and inhibits NK cell activity in vitro, as demonstrated in Figure 2b. This was evaluated using 1) YTS cells, which naturally do not express TIGIT (YTS Eco), and in which blocking TIGIT does not have any impact 2) YTS-TIGIT cells in which the authors demonstrate that *C. albicans* proteins bind to the TIGIT receptor, resulting in the inhibition of the cellular antifungal activity. In contrast to YTS eco, the addition of α -TIGIT antibody increased (restored) the antifungal activity of YTS-TIGIT cells. Nevertheless, it remains unclear why the anti-*Candida* activity is lower in YTS Eco cells not expressing TIGIT compared to that of YTS-TIGIT cells after addition of an α -TIGIT antibody. Does the α -TIGIT antibody exhibit activating properties? Does the TIGIT receptor have activating properties and does the binding of the α -TIGIT antibody boost the antifungal activity of the transduced (YTS-TIGIT) cells? From the data presented in Figure 2a one cannot conclude that TIGIT is "functional and inhibitory for NK cells" (page 7, line 151). Authors, please address these comments/questions.

Reviewer #1 raises two important issues:

1. Regarding the difference in anti-candidal activity between the YTS Eco and the YTS TIGIT cells. We also expected that the addition of anti-TIGIT to the YTS TIGIT cells would lead to similar anti-fungal activity as observed in YTS Eco cells. We think that the differences are due to clonality of the different YTS cell lines. After the transgene encoding for TIGIT was transduced into the parental YTS Eco cell line, a single TIGIT-positive clone was chosen and further propagated to establish the YTS TIGIT line. As a result, it might behave slightly different as compared to the parental YTS Eco cells and express differently other receptors (such as 2B4, see figure below) that are important for YTS killing.

Due to this reason, in fig. 2a we don't compare directly between the two lines, but only within each of the lines separately, and compare the isotype control to the anti-TIGIT antibody. We now discuss this point in more details in the manuscript (See page 7).

2. Regarding the reviewer's comment that "From the data presented in Figure 2a one cannot conclude that TIGIT is "functional and inhibitory for NK cells" (page 7, line 151)".

We completely agree. We based our conclusion on figure 2b which shows the effect of TIGIT on the anti-fungal activity of primary NK cells obtained from several human donors. Following these comments, we have edited the text to emphasize and clarify these points (Pages 7-8).

2. The y-axes in Figure 3b show different ranges, which means that the antifungal activity of the different mutants is different. Authors, please adjust the y-axes to allow a better comparison of the mutants and explain these differences.

We thank the reviewer for this comment. The relevant axes were adjusted accordingly.

It is now easier to observe that there isn't a major difference between the antifungal activities against the various mutants. The minor differences that are still present are due to the fact that each experiment shown in figure 3b was performed using primary NK cells isolated from different individuals. As the reviewer knows each primary NK cell line expresses NK activating and inhibitory receptors differently and thus have a different basic activity level, not only against fungi but also against various other tumor targets. Importantly however, when each NK line is compared to itself, it can be seen that they all respond to TIGIT blockade unless the Candida strain used is missing one of the TIGIT ligands.

3. The authors report that NK cells express TIGIT, which binds Als6/7/9 of *C. albicans* as ligands. The binding of these proteins (Als6/7/9) leads to the inhibition of NK cell activity, both in vitro and in vivo. α -TIGIT antibody counteracts this binding and therefore also NK cell inhibition by Als6/7/9, resulting in increased antifungal activity of NK cells. Mutant *C. albicans* strains lacking als 6, 7, or 9 (als6/7/9 Δ/Δ) do not inhibit NK cell activity, which demonstrates the importance of these fungal proteins in decreasing NK cell activity. In Figure 6c, it remains unclear why als6 Δ/Δ *C. albicans* and als7 Δ/Δ *C. albicans* (still expressing Als7 and 9 or Als6 and 9, respectively) do not inhibit NK cells, whereas als9 Δ/Δ (still expressing Als6 and 7) exhibits NK cell inhibition in vivo (Figure 6c), resulting in high mortality in *C. albicans* infected mice. Notably, the mortality is not significantly different from that of the wild type *C. albicans*. In addition, what is the impact on antifungal activity and mortality if combining different mutants (e.g., als6 Δ/Δ and als7 Δ/Δ or als6 Δ/Δ and als9 Δ/Δ etc)?

1. We have also wondered about the different in virulence observed between the mutant

strains for the Als6, Als7, and Als9 proteins. The Als proteins are well-known virulence factors playing an important role in adhesion. As each member of the Als family is expressed in different levels at different infection sites and times, it is hypothesized that each one fulfills different functions during *C. albicans* invasion. As such, it was previously demonstrated that each Als member can have several functions and several ligands (Hoyer, L. L. and Cota, E. (2016) 'Candida albicans agglutinin-like sequence (Als) family vignettes: A review of als protein structure and function', *Frontiers in Microbiology*), (Klotz, S. A. et al. (2004) 'Degenerate Peptide Recognition by Candida albicans Adhesins Als5p and Als1p', *Infection and Immunity*). Thus, while both Als6, Als7, and Als9 bind TIGIT, each one of them also binds different other proteins. As a result, deletion of each Als protein leads to a different effect on virulence.

Under the conditions used in our experiment (CFU injected per mouse, the specific mouse strain, and for 6a [6b after this revision] the specific organ examined and time of examination, etc.) the role of Als6 and Als7 is more pronounced, as their deletion cripples the virulence of *C. albicans*. Deletion of Als9 did not lead to a significant change in virulence. This is probably due to the numerous effects of Als9 deletion; it prevents TIGIT activation, but also changes the adhesion properties of the cells, changing their organ tropism in the mouse and the clinical disease the fungus causes. This leads to a complex phenotype, in which the lack of ability to activate TIGIT is probably balanced with the different adhesion and tissue tropism properties of the Als9 deletion mutant, resulting in an equal virulence to the WT strain. Similar effects have been shown for other Als proteins, that when deleted can in some conditions paradoxically increase adhesion (Zhao, X., Oh, S. H. and Hoyer, L. L. (2007) 'Deletion of ALS5, ALS6 or ALS7 increases adhesion of Candida albicans to human vascular endothelial and buccal epithelial cells', *Medical Mycology*).

Therefore, to differentiate between the various functions of the Als proteins and their effect on TIGIT, we compared between mock treatment and TIGIT blockade of each of the deletion mutants separately (in 6a and 6b [6b and 6c after revision]). This enabled us to observe that deletion of each of the Als proteins led to a loss of effect of the anti-TIGIT treatment, even though the background effect of the deletion is different for each Als gene deleted.

As we think this point that the reviewer brought up is indeed important for understanding of our results and was not explained clearly enough in the original text. We now better explain it in the revised manuscript (see pages 14-15).

2. The idea of combining different Als deletion mutants for the in vivo infections is indeed an interesting one. We also considered performing such an experiment but have decided against it due to several reasons. First, it can be seen that deletion of each of the single Als proteins that act as a TIGIT ligand is enough to completely abrogate the effect of TIGIT blockade. Co-infection of the mice with a combination of strains, or creating new strains deleted for several Als proteins, will most likely still maintain the observed complete elimination of this effect, and so will not fundamentally change our understanding of the reported phenomenon. Therefore, it is considered as unethical and will not be approved by our ethical committee.

Reviewer #1's question raises another interesting issue; why a deletion of each single Als member is enough to obtain a full effect? We think this is an important and interesting issue, and we now discuss it in our discussion section. Our main hypothesis, which was also raised by others (Hoyer, L. L. and Cota, E. (2016) 'Candida albicans agglutinin-like sequence (Als) family vignettes: A review of als protein structure and function', *Frontiers in Microbiology*), is that the Als proteins work as a complex. While all three members can independently bind TIGIT (as shown in figures 3+4), they are non-redundant and each one plays a critical role. Another option that was also suggested previously (Hoyer, L. L. and Cota, E. (2016) 'Candida albicans agglutinin-like

sequence (Als) family vignettes: A review of als protein structure and function', *Frontiers in Microbiology*) is that Als proteins regulate the expression levels of other members of the family, and the deletion of one lead to significant changes in expression of the others. We now discuss this in the discussion section (see page 16).

4. The authors state in the discussion section (page 14, line 297-298) that "deletion of just one of them [Als6, 7 or 9] is enough to completely abolish their effect on TIGIT" (page 14, line 297-298). However, data from Figure 6b and 6c show that deletion of als9 does not protect the mice from *C. albicans* infection, since the survival curve of the mice is similar of that of WT *C. albicans*, and therefore, this finding does not support this statement.

This important point Reviewer #1 raises is linked to our answer for the previous point. What we meant in the abovementioned sentence is that the deletion of each Als member (Als6, 7 or 9) is enough to abrogate the effect of TIGIT blockade, but as each Als member also affects virulence, their deletion leads to other effects unrelated to TIGIT activation. Due to these other effects, deletion of Als6 and Als7 leads to a more prominent reduction in virulence relative to Als9 deletion. This is why we compare each strain to itself with and without TIGIT blockade in figure 6. As mentioned in the previous point, this point was not explained well. We now better explain it in the results and discussion sections (see page 14-16).

Reviewer #2 (Remarks to the Author):

The manuscript by Yoav Charpak-Amikam and colleagues aims to describe a new fungal immune evasion step of the human pathogenic yeast *Candida albicans*.

The authors approach how *Candida albicans* can inactivate and dampen the immune attack of human natural killer cells. The group focuses on the immune inhibitory receptor expressed on the surface of human NK cells, TIGIT, and they use a screening approach to identify the fungal ligands. Upon screening of a fungal deletion library, the authors identify candida Als family proteins as ligands for both human and mouse TIGIT receptors. In addition, they used an in vivo infection model where they approach the role of several Als isoforms, and furthermore they use a monoclonal antibody to block Als protein function.

This concept is of interest and may ultimately show a new avenue of fungal and in particular candida immune evasion. However, in the present form, additional work is required to clearly and unambiguously confirm the concept presented by the authors. Numerous additional controls experiments are necessary and need to be presented in order to provide a clear evidence that candida Als proteins interact with the immune inhibitory receptor TIGIT expressed on human and mouse NK, as well as T cells. In addition further proof is necessary to conclude that the TIGIT -Als interaction is the central avenue of fungal immune escape.

We thank reviewer #2 for the interesting and constructive review. Following reviewer 1 and 2 comments 6 new figures were added (fig. 2c,d,e, 3e and S1c-d) and 5 more were modified and improved (fig. 1d, 3b, 4 and 5a-b). We have amended the manuscript in accordance with the following comments.

In addition, regarding the reviewer's last sentence dealing with the TIGIT-Als interaction as the central avenue of fungal immune escape, we wish to emphasize that our claim is less dramatic and more specific. Fungal pathogens are many and varied, each having its own immune evasion strategy. *Candida albicans* evolved many immune evasion mechanisms,

some known and some yet to be discovered. We think that the interaction with TIGIT is indeed an important immune evasion mechanism, and it is definitely novel since the fungus hijacks immune checkpoint receptors for its needs, but we do not claim it is the central avenue for immune evasion.

Issues in details:

1. Als has several family members. Does each Als protein bind to the TIGIT receptors with similar affinity and is this interaction specific for human NK cells. What is the common binding motive.

The issues reviewer #2 raises here are indeed important to the understanding of the biology behind the phenomenon we describe in our manuscript.

1. The Als protein family indeed contains many members, each having a different functional role and different ligands. To evaluate the binding affinities of each TIGIT-Als protein interaction we cloned them, produced recombinant versions of these proteins, and used both FACS and MST to test whether they bind to TIGIT and with what affinity (fig. 3c-d and the new fig. 4). Using MST we observed different binding affinities: with Als6 being the strongest TIGIT binder and Als9 being the weakest (new figure 4). This is also in line with the BW experiment (fig. 3a) that shows a more pronounced reduction in TIGIT activation for the Als6-deleted candida strain relative to the Als9-deleted strain (even though they both activate TIGIT significantly less than the WT strain).

2. This described interaction is not specific to human NK cells, as a recombinant TIGIT protein from mouse was also able to bind to *C. albicans* cells (fig. 5c-d), and deletion of Als6, Als7 or Als9 was able to abrogate the effects of TIGIT blockade during in vivo infections of mice (fig. 6).

3. Als proteins are usually composed of several domains. When cloning the Als-Ig fusion proteins used in figures 3+4 we only used the N-terminal domain. This was due to technical reasons (additional domains present in Als proteins have high potential for aggregation) and also due to functional reasons; the N-terminal domain is known to be the main ligand-binding domain for Als proteins and contains the motif which is responsible for binding all of the known Als ligands. This was determined using both functional data and structural data that is available for several Als proteins including Als9-2 (Salgado, P. S., Yan, R., Rowan, F., *et al.* (2011) 'Expression, crystallization and preliminary X-ray data analysis of NT-Als9-2, a fungal adhesin from *Candida albicans*', *Acta Crystallographica Section F: Structural Biology and Crystallization Communications*), (Salgado, P. S., Yan, R., Taylor, J. D., *et al.* (2011) 'Structural basis for the broad specificity to host-cell ligands by the pathogenic fungus *Candida albicans*', *Proceedings of the National Academy of Sciences*).

As reviewer #2 raises an interesting and important point that was not emphasized properly in the text, we edited the relevant parts in the text to further elaborate on it (see pages 11-12 and 16).

2. Do all members of the ALS family proteins bind to TIGIT.

In order to check which Als family members bind TIGIT we examined deletion mutants for each and every Als gene in *C. albicans* using a Bw assay and found that only Als6, Als7 and Als9 activate TIGIT (fig. 3a). We also cloned recombinant versions of these proteins to validate and quantify this finding, which led us to also determine that only one allele of Als9, Als9-2, but not Als9-1, is a TIGIT ligand.

3. Also T cells express TIGIT receptors, what is the effect of the candida protein's on these central type of immune human cells.

We thank reviewer #2 for this comment. T cells indeed play a central role in anti-fungal immunity. To test whether *Candida albicans* uses TIGIT activation to inhibit T cells in addition to NK cells we performed the experiment described in new figures 2c+d. In short, we isolated primary CD4⁺TIGIT⁺ T cells from the blood of human donors, activated them, and checked whether *Candida albicans* cells inhibit them, and whether this inhibition can be abrogated using an anti-TIGIT antibody. We saw that TIGIT blockade could indeed increase the activation levels of T cells (new figure 2d).

In addition, we also examined the role of T cells in our in-vivo invasive candidiasis model (figures 5a+b) and observed a significant role for T cells as well.

These new results lead us to suggest that the TIGIT-mediated immune evasion of *Candida albicans* works via inhibition of not only NK cells, but also T cells. We now also discuss this in the results and discussion sections (see pages 8, 12-13 and 15-16).

4. Does the mAB used here block binding of the recombinant TIGIT to candida, also block the binding of each Als protein to TIGIT in the same manner. Does the antibody show the same inhibitory effect for all Als proteins.

The ability of the anti-TIGIT antibody we use here to block TIGIT was indeed only indirectly shown (in figure 3b for example). In order to directly show the ability of the antibody to block the interaction with the Als proteins and to test whether it has the same effect on all TIGIT-Als interactions, we repeated the experiment depicted in figure 3c+3d (staining of TIGIT-expressing BW cells with recombinant Als-Ig fusion proteins), in the presence of the anti-TIGIT antibody. As can be seen in new figure 3e, the antibody indeed blocks these interactions.

Regarding the strength of the inhibitory effect, figure 3e show that the anti-TIGIT antibody has more or less the same effect on all NT-Als-Ig fusion proteins. Figure 3b shows that this effect is also similar functionally, as deletions of either Als protein completely abolished the effects of TIGIT blockade on the *Candida* cells' immune evasion abilities.

5. The authors use fusion proteins in most experiments, they should include additional assays in which native untagged proteins are used, e.g. for immunoprecipitation.

Many experiments in the manuscript indeed include fusion proteins that include an Ig-tag to identify and quantify them. This tag is also critical in the generation of these proteins as it is used in the protein G affinity column, leading to very high purity (near 100%). The use of untagged proteins is indeed more favorable and can prevent some experimental artifacts. We wished to perform such experiments, but unfortunately, to the best of our knowledge no antibodies against Als6, Als7 or Als9 are available commercially or described in the literature. We also attempted to manufacture these proteins using other tags, but were unsuccessful.

With that in mind, we tried to control for the presence of the Ig tag using several methods:

1. We used another reagent, the *Candida* Als deletion mutants, to examine the interaction between Als proteins and TIGIT independently of the tagged Als proteins (fig. 3a+b and fig. 6).

2. We performed MST experiments using only TIGIT and Als recombinant proteins to evaluate

the Als-TIGIT interaction directly and exclude possible artifacts originating from the use of antibodies or from additional variables that might be present on human or candida cells (please see new figure 4).

3. We used a control fusion protein containing an identical Ig tag, produced in the exact same manner as a negative control in all experiments that included the NT-Als-Ig fusion proteins.

4. In addition to our extensive experience with Ig-tagged proteins (We have used them in many published projects over many years, for example (Mandelboim, O. *et al.* (2001) 'Recognition of haemagglutinins on virus-infected cells by NKp46 activates lysis by human NK cells.', *Nature*), (Stanietsky, N. *et al.* (2009) 'The interaction of TIGIT with PVR and PVRL2 inhibits human NK cell cytotoxicity.', *Proceedings of the National Academy of Sciences of the United States of America*), (Vitenshtein, A. *et al.* (2016) 'NK Cell Recognition of *Candida glabrata* through Binding of NKp46 and NCR1 to Fungal Ligands Epa1, Epa6, and Epa7', *Cell Host and Microbe*), (Reches, A. *et al.* (2020) 'Nectin4 is a novel TIGIT ligand which combines checkpoint inhibition and tumor specificity', *Journal for Immunotherapy of Cancer*), we also went over the literature (as this tag is also widely used in many other labs) and did not find evidence for possible artifacts or effects on the tagged proteins that could be relevant to our study.

6. Is this a uniform mechanism of fungal Als function? Do other fungi, like *Aspergillus fumigatus* use the same type of immune escape. Do *Aspergillus fumigatus* Als proteins, bind to the same human and mouse TIGIT ligands and do they have the same biological effects.

Reviewer #2 raises an interesting question, concerning both the biological mechanism of the interaction we describe and the biological scope of this finding.

In order to answer this question, we performed another experiment described in new fig. 1d and tested whether TIGIT-Ig binds to non-*albicans* candida species. To further emphasize this point, we now used pathogenic fungi that are not candida; *Cryptococcus neoformans* and *Cryptococcus gattii*. These two species were not recognized by TIGIT-Ig (new fig 1d). This strengthens our conclusion that this immune evasion mechanism is species-specific and not all pathogenic fungi has it.

This finding is also in line with our recognition of Als proteins as the TIGIT ligands, as these proteins are very species-specific and are not present in non-candida pathogenic fungi (Hoyer, L. L. and Cota, E. (2016) 'Candida albicans agglutinin-like sequence (Als) family vignettes: A review of als protein structure and function', *Frontiers in Microbiology*). Moreover, even within the Candida family not all species encode Als genes, and only the ones that are known to contain Als (*Candida albicans* and *Candida parapsilosis* (Oh, S.-H. *et al.* (2019) 'Agglutinin-Like Sequence (ALS) Genes in the *Candida parapsilosis* Species Complex: Blurring the Boundaries Between Gene Families That Encode Cell-Wall Proteins', *Frontiers in Microbiology*) were able to bind TIGIT. We now also discuss this in the discussion section (see page 17).

7. What is the signaling pathway initiated by Als proteins and what is the cell response. Is the effect the same in NK cells and in T lymphocytes? The immune dampening effect need to be evaluated in greater detail.

The intracellular signaling cascade following TIGIT activation is indeed an interesting point

that we did not touch in the original version of the manuscript. The signaling cascade initiated by the binding of TIGIT to its endogenous ligands was extensively examined in the past, by both our lab and others (Stanietsky, N. *et al.* (2009) 'The interaction of TIGIT with PVR and PVRL2 inhibits human NK cell cytotoxicity.', *Proceedings of the National Academy of Sciences of the United States of America*), (Liu, S. *et al.* (2013) 'Recruitment of Grb2 and SHIP1 by the ITT-like motif of TIGIT suppresses granule polarization and cytotoxicity of NK cells', *Cell Death and Differentiation*). This signaling cascade was mostly studied in NK cells, and while it is less studied in T cells, it is hypothesized to be similar in its initial stages. This signaling cascade includes phosphorylation of critical tyrosine residues in the cytoplasmic tail of the protein, recruitment of adapter proteins and intracellular phosphatases, and dampening of signals produced by various kinases.

In order to examine whether the signaling cascade initiated by Candida activation of TIGIT is similar to the one described for the endogenous ligands we used TIGIT-expressing YTS cells mutated in the ITIM domain which is critical for the initiation of the signaling cascade. We used two mutants: one with a point mutation in the ITIM's tyrosine residue (YTS TIGIT Y231A), and one with a stop codon interrupting the ITIM and the final 13 amino acids following it (YTS TIGIT Y231Stop). This model was used in the past by our lab and others to characterize the signaling events responsible for the inhibitory effects of TIGIT (Stanietsky, N. *et al.* (2009) 'The interaction of TIGIT with PVR and PVRL2 inhibits human NK cell cytotoxicity.', *Proceedings of the National Academy of Sciences of the United States of America*), (Liu, S. *et al.* (2013) 'Recruitment of Grb2 and SHIP1 by the ITT-like motif of TIGIT suppresses granule polarization and cytotoxicity of NK cells', *Cell Death and Differentiation*). As can be seen in new figure 2e, any interference with the ITIM motif completely abrogated the Candida-mediated TIGIT inhibition. We now also discuss this in the results and the discussion sections (see pages 8-9 and 17).

8. Also in this regard the Als-TIGIT Interaction may be relevant for mediated NK cell activity. But the effect on T cells needs to be included. In this regard, if the concept of Als-TIGIT interaction is really important than the effect on the host immune response mediated by NK and by T cells needs to be explored

We completely agree with reviewer #2. As mentioned in point #3, we have added both in vitro and in vivo data regarding the effects of the Candida-TIGIT interaction (please see new figures 2c-d and 5a-b).

9. Figure 1. One problem in this figure is presented in panels 1b, 1c, and 1d by the presentation of relative values. What are the real binding data. Like in 1c, a value two times about control may not be a dramatic high affinity binding.

We thank reviewer #2 for this important comment. We also agree that presentation of raw data together with relative values is important in order to report the data in a way that combines and maximizes both transparency and comprehensibility. To that end, we included a histogram showing representative raw data with most experiments (for example Fig. 1a, 3c and 5c).

To provide the reviewer with the requested values we bring below graphs showing the raw data corresponding to figures 1b, 1c and 1d. One of the main reasons we choose to show quantifications of multiple FACS and ELISA experiments using data relative to a constant control is due to high inter-experimental changes occurring due to various technical issues such as using different machines, different settings, different experimenters and so on. This

can also be observed in the raw data presented below. To present the data accurately we changed the Y-axis from linear to logarithmic in the following graphs (in the FACS experiments). In addition, one specific data point in fig. 1b is significantly different from the rest, so we present the data with and without it (fig. 1b1 vs 1b2).

10. For panels 1e and 1f. Whether controls include like using the mAb to block the interaction. How is the effect on binding to the various als knock out mutants. In particular given the redundancy of the various Als proteins this aspect needs to be addressed .

Panels 1e and 1f present an immunofluorescence microscopy experiment in which we stained yeast and hyphae candida cells to understand whether the binding we observed between TIGIT and candida yeast cells is also present in other morphologies of *C. albicans*. Due to that, we only performed controls necessary for the technical aspect of experiment (such as using a control-Ig protein as a negative control and staining yeast cells as a positive control). We agree with reviewer #2 that additional characterization of this interaction, including the specific effects of the different Als ligands or the blocking antibody we use in experiments in later parts of the manuscript are indeed important. We tried using confocal microscopy to characterize and quantify these interactions, but due to the limitation in quantification inherent in the method, we did not succeed, although a trend was observed. Therefore, we prefer showing more reliable quantitative assays, such as FACS, ELISA, BW and NK cytotoxicity (fig. 3a,b,c,d for the effect of the different Als ligands and fig. 2a, 2b, 2d and 3e for the effects of the antibody). If reviewer #2 still feels that the microscopy experiment presented in fig. 1e-f is missing critical controls, we will repeat the confocal experiments many times to obtain statistical differences.

11. Figure 2. What is the meaning of the clinical strains. The clinical strains are an impressive selection. It would be nice to compare the expression levels of the various Als isoforms at least in several of the high and low effective isolates.

The mechanism behind the differential ability of the clinical *C. albicans* isolates to activate TIGIT is indeed an interesting question. We checked the Als mRNA expression levels of the five most activating isolates and the three weakest activating isolates using quantitative real-time PCR (please see figure below). As controls we also measured the Als expression levels in the WT lab strain (SC5314) and its three daughter strains deleted for Als6, Als7 or Als9.

As can be seen in the figure, we did not identify any significant differences. This suggests that another mechanism, most likely a post-transcriptional one, is responsible for the difference in TIGIT activation abilities in the examined strains. Such differences can be on the protein level, for example differences in translation efficiencies, protein degradation rates, protein modifications, protein localization, and so on. Another possible might be that the Als proteins work as a complex and could be regulating the activity and/or expression levels of each other, creating a complex regulatory network between themselves (Hoyer, L. L. and Cota, E. (2016) 'Candida albicans agglutinin-like sequence (Als) family vignettes: A review of als protein structure and function', *Frontiers in Microbiology*).

Unfortunately, to the best of our knowledge no antibodies are available that directly bind Als6, Als7, Als9-1 or Als9-2. As such, probing differences on the protein level is highly complicated. In addition, it is likely that each clinical isolate evolved a different mechanism to change its Als protein activity. We now discuss this in the discussion section (see page 17).

12. Figure 3. The effect TIGIT activation her all nine als knock out mutant were assessed What is the contribution of each als isoform on expression level and on TIGIT mediated activation. Please show also the raw data and not only the relative effects.

The aforementioned experiment is complex. For example, we realized that TIGIT activation levels are affected by the specific growth conditions of the BW cells, and that the live Candida cells affect the BW cells and their ability to be activated and secrete IL-2, probably due to the sensing of fungal metabolites and the secretion of fungal proteases, etc..

Due to that, we analyzed the experiments' results using three stages.

First, we perform the co-incubation of the Candida cells with either parental BW cells or TIGIT-expressing BW cells. BW cell activation is measured by the secretion of IL-2 (using anti-IL-2 ELISA).

Next, we normalize the results from each BW TIGIT x Candida sample to an identical reaction with the same Candida strain in the presence of the BW parental cells. This allows us to better control for the various non-TIGIT related effects and artifacts.

Finally, we normalize these results to the measured values of the BW TIGIT x WT Candida in each experiment, to control for inter-experimental variability (which tends to be quite high sometimes).

From our experience this experimental scheme provides accurate and better controlled results. Because of the reviewer's comment, we understood that this process isn't explained well enough in the manuscript, so we edited and improved it (please see the relevant methods sub-section, pages 28-29). In addition, we provide below the raw data obtained from the different stages:

Data analysis stages for the BW activation assay. TIGIT activation was assayed using the murine thymoma cell line BW expressing a chimeric TIGIT-z-chain receptor. The BW cells were co-incubated for 48 hours in the presence or absence of the WT *C. albicans* strain SC5314 or mutant strains deleted for members of the Als protein family. Identical wells containing parental BW cells not expressing TIGIT were also prepared for each experimental condition. TIGIT activation was measured using ELISA for IL-2 secreted by the activated BW cells. Shown are averages of 2-4 independent experiments. **The upper figure** shows the raw OD values read from the ELISA plates. **The middle figure** shows these values after normalization to their matched parental BW samples for better control of various internal experiment noise. **The lower figure** shows the final data after normalization to the WT candida strain in each experiment, in order to better control for noise and technical differences between different experiments.

13. Figure 4. The effect a control protein (PVR-Ig) is comparable to Als7 protein. Does this

reflect a specific interaction and high affinity binding? As negative control and lacking IgG domain needs to be included. In addition, k_{on} and k_{off} values and rates need to be presented. A critical interpretation of these results is that the also Ig fusion proteins show related binding at the human ligand PVR Ig. However, a negative control needs to be presented.

1. Indeed, the MST results for the interaction between TIGIT and both NT-Als7-Ig and PVR-Ig are very similar (40-50nm), with the binding affinity for NT-Als6-Ig even stronger (6nm) and the affinity for NT-Als9-2-Ig weaker (455nm). All these interactions are within an order of magnitude lower than our positive control; the well characterized interaction between TIGIT and its endogenous ligand PVR.

As such, we conclude that all these interactions are biologically relevant. Additional support for this conclusion is the fact that the binding affinities between the Als-Ig fusion proteins and TIGIT are similar to those observed in the BW experiment (fig. 3a). Deletion of Als6 leads to the strongest effect on TIGIT activation and deletion of Als9 leads to the weakest. We have also edited the relevant part in the text to better clarify and emphasize these points (see page 12).

2. We agree that a negative control should be included. A negative control was of course included in the original experiment but not shown. This is fixed now, and the new figure 4 includes it.

3. Potential artifacts originating from the IgG Fc domain were controlled by using a negative control protein that includes an Ig tag, and now appears in the new figure 4 and the text (see page 12). In addition, we feel confident that it does not significantly affect the interaction with TIGIT-Ig as we have performed several similar MST experiments with various ligands of TIGIT fused to this tag in different published projects over the years (for example in (Rechtes, A. *et al.* (2020) 'Nectin4 is a novel TIGIT ligand which combines checkpoint inhibition and tumor specificity', *Journal for Immunotherapy of Cancer*).

4. Unfortunately, K_{on} and K_{off} values are not possible to get when using MST (this is a known limitation of the instrument).

14. Figure 5. The effect Figure 5a and 5b:

Figure 5d again please present the experimental data without the calculation of the value relative to control. What is the effect in T cell depleted animals.

1. New T cell data was added to new figures 5a and 5b, showing that T cells are also involved in the control of invasive candidiasis. In accordance with these results we added further discussion of this subject to various sections of the manuscript (see pages 8-9 and 15-16).

2. We agree with the reviewer that it is important to take into account both the raw data and the relative one to properly assess the reported effect. As mentioned in a previous comment, we initially added the representative histogram (fig. 5c), in addition to the quantification of the relative results (fig. 5d). We think that the combination of the two data formats provides the best balance between data transparency and comprehensibility.

In addition, we show below a quantification of the raw data without normalization to the control-Ig. Please notice that the Y-axis was converted into log scale in order to better present the data. This is due to high variability in the signal strength between the experiments (due to using different FACS machines at different times using slightly different technical settings) and

can be an example for our reasoning behind presenting relative data in addition to the histogram presenting the raw data.

15. Figure 6.

The effect of the various *C. albicans* Als knock out mutant in the in vivo infection model need to be evaluated and discussed first. Then the role of mAB can be addressed first.

What is the reason on selecting got the three mutants, als6, als7 and als9. Additional control like revertant strains for each knock out mutant studied need to be presented.

1. We re-ordered the figure and its corresponding parts in the manuscript accordingly, as suggested (see pages 13-14).

2. We chose to select the three Als mutants, *als6Δ/Δ*, *als7Δ/Δ*, and *als9Δ/Δ* for further in-vivo evaluation as the experiments depicted in figures 3 and 4 indicate that they are the major ligands of TIGIT present on *C. albicans*. We thought that in vivo experiments comparing the effects of TIGIT blockade on the mutants and WT strains will further validate their role as TIGIT ligands under physiological conditions. Another reason behind their inclusion was that they also serve as a control for the possible side effects of TIGIT blockade during invasive candidiasis. It is possible that TIGIT blockade can unspecifically disinhibit the immune response, as seen as a common complication of immune checkpoint inhibition in human patients. If that were to happen in our model, we would have expected to see the same effect during TIGIT blockade in all treated mice, independent of the candida strain infecting them. On the other hand, if the effects of the anti-TIGIT antibody are specific to the Als immune evasion phenomenon, and are not due to the potential general immune-disinhibitory effect of checkpoint inhibitors, the usage of the ligand-missing mutants would abrogate this effect. We edited the text to better describe these points (see pages 13-15).

3. Adding revertants is indeed a nice suggestion but it require a very long time to complete these experiments (in our opinion more than a year, as we have to generate them and then test them using the various in vitro and in vivo assays). As we are already working on this project for around five years, we hope that the reviewer will agree to the current manuscript being published without the revertants. We will work on them in the future related projects.

16. Language: The text needs revision in language and also in content. Some of the statements sound rather dramatic and in the end are not absolutely right.

E.g. The point 'that *Candida albicans* is ... one of the deadliest causes of bloodstream infections' sounds very dramatic but is not complement right (page 2, abstract first sentence). Do the authors really want to say that other pathogenic bacteria like

Streptococcus pneumoniae, S pyogenes, Staphylococcus aureus are not relevant. In particular not to mention the current COVID-19 situation which gives another example on severity of blood stream infections.

We thank the reviewer for this critical point as we always wish to report our data in an accurate and leveled fashion. As such, we went over the text and modified the relevant statements (see page 2 and additional statements throughout the paper).

Regarding the specific point the reviewer raised, we referred to the data presented in the introduction showing that *C. albicans* infection is extremely common, being the 3rd to 4th most common cause of bloodstream infection in hospitalized patients in many countries (and being responsible for 18-22% of such infections in the US). It is also extremely deadly, as 40-60% of all infected patients die (Pappas, P. G. *et al.* (2018) 'Invasive candidiasis', *Nature Reviews Disease Primers*). We did not, by any means, try to imply that other pathogens, whether they be bacteria such as Staphylococci or Streptococci, viruses such as SARS-COV-2, or other fungi or even parasites, are less relevant.

Unfortunately for patients, all of these pathogens are still equally deadly and important and lead to high morbidity and mortality. The case of COVID-19 which the reviewer brought up is of specific interest, not only due to the global pandemic, but also due to the fact that like most intensive care admitted patients (and especially those treated with steroids), individuals suffering from COVID-19 are at a very high risk of fungal bloodstream infections which takes the lives of many of them.

In order to prevent further misunderstandings regarding this statement we amended and softened it, and went over the text to amend similar statements.

REVIEWERS' COMMENTS

Reviewer #1 (Remarks to the Author):

Reviewer #1 (Remarks to the Author):

Charpak-Amikam et al report on the identification of the TIGIT receptor as the NK cell receptor binding to *Candida albicans* yeast and hyphae. Via this receptor, NK cells recognize the *C. albicans* cell wall-bound adhesion proteins Als6, Als7 and Als9. The ligands binding to this receptor lead to inhibition NK cell function, and in vivo studies in mice infected with *C. albicans* demonstrate higher mortality. As fungal infections are still a major cause of morbidity and mortality in different patient populations, the present data are of high interest for a better understanding of the host-fungal interaction and may provide the rationale for new therapeutic strategies.

Although the data are well presented and quite convincing, I would like to make several comments.

We thank reviewer #1 for the thoughtful and interesting review. Following reviewer 1 and 2 comments 6 new figures were added (fig. 2c,d,e, 3e and S1c-d) and 5 more were modified and improved (fig. 1d, 3b, 4 and 5a-b). We have amended the manuscript in accordance with the following comments:

1. The authors state that *C. albicans* binds NK cell TIGIT and inhibits NK cell activity in vitro, as demonstrated in Figure 2b. This was evaluated using 1) YTS cells, which naturally do not express TIGIT (YTS Eco), and in which blocking TIGIT does not have any impact 2) YTS-TIGIT cells in which the authors demonstrate that *C. albicans* proteins bind to the TIGIT receptor, resulting in the inhibition of the cellular antifungal activity. In contrast to YTS Eco, the addition of α -TIGIT antibody increased (restored) the antifungal activity of YTS-TIGIT cells. Nevertheless, it remains unclear why the anti-*Candida* activity is lower in YTS Eco cells not expressing TIGIT compared to that of YTS-TIGIT cells after addition of an α -TIGIT antibody. Does the α -TIGIT antibody exhibit activating properties? Does the TIGIT receptor have activating properties and does the binding of the α -TIGIT antibody boost the antifungal activity of the transduced (YTS-TIGIT) cells? From the data presented in Figure 2a one cannot conclude that TIGIT is “functional and inhibitory for NK cells” (page 7, line 151). Authors, please address these comments/questions.

Reviewer #1 raises two important issues:

1. Regarding the difference in anti-candidal activity between the YTS Eco and the YTS TIGIT cells. We also expected that the addition of anti-TIGIT to the YTS TIGIT cells would lead to similar anti-fungal activity as observed in YTS Eco cells. We think that the differences are due to clonality of the different YTS cell lines. After the transgene encoding for TIGIT was transduced into the parental YTS Eco cell line, a single TIGIT-positive clone was chosen and further propagated to establish the YTS TIGIT line. As a result, it might behave slightly different as compared to the parental YTS Eco cells and express differently other receptors (such as 2B4, see figure below) that are important for YTS killing.

Due to this reason, in fig. 2a we don't compare directly between the two lines, but only within each of the lines separately, and compare the isotype control to the anti-TIGIT antibody. We now discuss this point in more details in the manuscript (See page 7).

2. Regarding the reviewer's comment that "From the data presented in Figure 2a one cannot conclude that TIGIT is "functional and inhibitory for NK cells" (page 7, line 151)".

We completely agree. We based our conclusion on figure 2b which shows the effect of TIGIT on the anti-fungal activity of primary NK cells obtained from several human donors. Following these comments, we have edited the text to emphasize and clarify these points (Pages 7-8).

Changes accepted - no further comment.

2. The y-axes in Figure 3b show different ranges, which means that the antifungal activity of the different mutants is different. Authors, please adjust the y-axes to allow a better comparison of the mutants and explain these differences.

We thank the reviewer for this comment. The relevant axes were adjusted accordingly.

It is now easier to observe that there isn't a major difference between the antifungal activities against the various mutants. The minor differences that are still present are due to the fact that each experiment shown in figure 3b was performed using primary NK cells isolated from different individuals. As the reviewer knows each primary NK cell line expresses NK activating and inhibitory receptors differently and thus have a different basic activity level, not only against fungi but also against various other tumor targets. Importantly however, when each NK line is compared to itself, it can be seen that they all respond to TIGIT blockade unless the Candida strain used is missing one of the TIGIT ligands.

Changes accepted. The adjustment of the axes makes it easier to compare the data and I appreciate that the authors followed my suggestion.

3. The authors report that NK cells express TIGIT, which binds Als6/7/9 of *C. albicans* as ligands. The binding of these proteins (Als6/7/9) leads to the inhibition of NK cell activity, both in vitro and in vivo. α -TIGIT antibody counteracts this binding and therefore also NK cell inhibition by Als6/7/9, resulting in increased antifungal activity of NK cells. Mutant *C. albicans* strains lacking als 6, 7, or 9 (als6/7/9 Δ/Δ) do not inhibit NK cell activity, which demonstrates the importance of these fungal proteins in decreasing NK cell activity. In Figure 6c, it remains unclear why als6 Δ/Δ *C. albicans* and als7 Δ/Δ *C. albicans* (still expressing Als7 and 9 or Als6 and 9, respectively) do not inhibit NK cells, whereas als9 Δ/Δ (still expressing Als6 and 7) exhibits NK cell inhibition in vivo (Figure 6c), resulting in high mortality in *C. albicans* infected mice. Notably, the mortality is not significantly different from that of the wild type *C. albicans*. In addition, what is the impact on antifungal activity and mortality if combining different mutants (e.g., als6 Δ/Δ and als7 Δ/Δ or als6 Δ/Δ and als9 Δ/Δ etc)?

1. We have also wondered about the different in virulence observed between the mutant strains for the Als6, Als7, and Als9 proteins. The Als proteins are well-known virulence factors playing an important role in adhesion. As each member of the Als family is expressed in different levels at different infection sites and times, it is hypothesized that each one fulfills different functions during *C. albicans* invasion. As such, it was previously demonstrated that each Als member can have several functions and several ligands (Hoyer, L. L. and Cota, E. (2016) 'Candida albicans agglutinin-like sequence (Als) family vignettes: A review of als protein structure and function', *Frontiers in Microbiology*), (Klotz, S. A. et al. (2004) 'Degenerate Peptide Recognition by Candida albicans Adhesins Als5p and Als1p', *Infection and Immunity*). Thus, while both Als6, Als7, and Als9 bind TIGIT, each one of them also bind different other proteins. As a result, deletion of each Als protein leads to a different effect on virulence.

Under the conditions used our experiment (CFU injected per mouse, the specific mouse strain, and for 6a [6b after this revision] the specific organ examined and time of examination, etc.) the role of Als6 and Als7 is more pronounced, as their deletion cripples the virulence of *C. albicans*. Deletion of Als9 did not lead to a significant change in virulence. This is probably due to the numerous effects of Als9 deletion; it prevents TIGIT activation, but also changes the adhesion properties of the cells, changing their organ tropism in the mouse and the clinical disease the fungus causes. This leads to a complex phenotype, in which the lack of ability to activate TIGIT is probably balanced with the different adhesion and tissue tropism properties of the Als9 deletion mutant, resulting in an equal virulence to the WT strain. Similar effects have been shown for other Als proteins, that when deleted can in some conditions paradoxically increase adhesion (Zhao, X., Oh, S. H. and Hoyer, L. L. (2007) 'Deletion of ALS5, ALS6 or ALS7 increases adhesion of Candida albicans to human vascular endothelial and buccal epithelial cells', *Medical Mycology*).

Therefore, to differentiate between the various functions of the Als proteins and their effect on TIGIT, we compared between mock treatment and TIGIT blockade of each of deletion mutant separately (in 6a and 6b [6b and 6c after revision]). This enabled us to observe that deletion of each of the Als proteins led to a loss of effect of the anti-TIGIT treatment, even though the background effect of the deletion is different for each Als gene deleted.

As we think this point that the reviewer brought up is indeed important for understanding of our results and was not explained clearly enough in the original text. We now better explain it in the revised manuscript (see pages 14-15).

2. The idea of combining different Als deletion mutants for the in vivo infections is indeed an interesting one. We also considered performing such an experiment but have decided against it due to several reasons. First, it can be seen that deletion of each of the single Als proteins that act as a TIGIT ligand is enough to completely abrogate the effect of TIGIT blockade. Co-infection of the mice with a combination of strains, or creating new strains deleted for several Als proteins, will most likely still maintain the observed complete elimination of this effect, and so will not fundamentally change our understanding of the reported phenomenon. Therefore, it is considered as unethical and will not be approved by our ethical committee.

Reviewer #1's question raises another interesting issue; why a deletion of each single Als member is enough to obtain a full effect? We think this is an important and interesting issue, and we now discuss it in our discussion section. Our main hypothesis, which was also raised by others (Hoyer, L. L. and Cota, E. (2016) 'Candida albicans agglutinin-like sequence (Als) family vignettes: A review of als protein structure and function', *Frontiers in Microbiology*), is that the Als proteins work as a complex. While all three members can independently bind TIGIT (as shown in figures 3+4), they are non-redundant and each one plays a critical role. Another option that was also suggested previously (Hoyer, L. L. and Cota, E. (2016) 'Candida albicans agglutinin-like sequence (Als) family vignettes: A review of als protein structure and function', *Frontiers in Microbiology*) is that Als proteins regulate the expression levels of other members of the family, and

the deletion of one lead to significant changes in expression of the others. We now discuss this in the discussion section (see page 16).

4. The authors state in the discussion section (page 14, line 297-298) that “deletion of just one of them [Als6, 7 or 9] is enough to completely abolish their effect on TIGIT” (page 14, line 297-298). However, data from Figure 6b and 6c show that deletion of *als9* does not protect the mice from *C. albicans* infection, since the survival curve of the mice is similar of that of WT *C. albicans*, and therefore, this finding does not support this statement.

This important point Reviewer #1 raises is linked to our answer for the previous point. What we meant in the abovementioned sentence is that the deletion of each Als member (Als6, 7 or 9) is enough to abrogate the effect of TIGIT blockade, but as each Als member also affects virulence, their deletion leads to other effects unrelated to TIGIT activation. Due to these other effects, deletion of Als6 and Als7 leads to a more prominent reduction in virulence relative to Als9 deletion. This is why we compare each strain to itself with and without TIGIT blockade in figure 6. As mentioned in the previous point, this point was not explained well. We now better explain it in the results and discussion sections (see page 14-16).

Points 3 & 4:

I agree with the author's explanation why not to combine different Als deletion mutants and Als proteins working as a complex. However, I still hesitate with the role of Als9. While the title of the according paragraph is “The Als-mediated immune evasion mechanism can be targeted using immunotherapy in mice”, the data demonstrated by the authors do not support Als9 as a target. The mouse experiment show that targeting Als9 has no effect on the virulence of the fungus and the survival of the animals. Although the deletion of *als9* had an effect in the cytotoxicity experiments in vitro (Fig 3b), this effect was not seen in vivo. In line 302-303 the authors state “... under our experimental conditions *als9Δ/Δ* appeared to not be significantly more pathogenic relative to the WT strain (Fig.6A)”. In addition, the authors state that “The *als9Δ/Δ* strain did lead to significant mortality and kidney fungal burden in the infected mice, and surprisingly did not show a significantly reduced pathogenicity relative to the WT strain.” (line 315ff). These facts suggest that blocking Als9 in vivo does not make a difference on the outcome of the infection. Similarly, “... the net effect [of] Als9 deletion, which consists of its pro- and anti-adhesion effects and its TIGIT activating abilities, is neutral.” (line 322). Therefore, I still wonder why Als9 should be a good target for immunotherapy.

Editorial Note: Reviewer #1 was asked to review the comments previously submitted by Reviewer #2, who was unable to review on this occasion.

This response is Reviewer Attachment #2.

Reviewer #2 (Remarks to the Author): The manuscript by Yoav Charpak-Amikam and colleagues aims to describe a new fungal immune evasion step of the human pathogenic yeast *Candida albicans*. The authors approach how *Candida albicans* can inactivate and dampen the immune attack of human natural killer cells. The group focuses on the immune inhibitory receptor expressed on the surface of human NK cells, TIGIT, and they use a screening approach to identify the fungal ligands. Upon screening of a fungal deletion library, the authors identify candida Als family proteins as ligands for both human and mouse TIGIT receptors. In addition, they used an in vivo infection model where they approach the role of several Als isoforms, and furthermore they use a monoclonal antibody to block Als protein function. This concept is of interest and may ultimately show a new avenue of fungal and in particular candida immune evasion. However, in the present form, additional work is required to clearly and unambiguously confirm the concept presented by the authors. Numerous additional

controls experiments are necessary and need to be presented in order to provide a clear evidence that candida Als proteins interact with the immune inhibitory receptor TIGIT expressed on human and mouse NK, as well as T cells. In addition further proof is necessary to conclude that the TIGIT -Als interaction is the central avenue of fungal immune escape.

We thank reviewer #2 for the interesting and constructive review. Following reviewer 1 and 2 comments 6 new figures were added (fig. 2c,d,e, 3e and S1c-d) and 5 more were modified and improved (fig. 1d, 3b, 4 and 5a-b). We have amended the manuscript in accordance with the following comments.

In addition, regarding the reviewer's last sentence dealing with the TIGIT-Als interaction as the central avenue of fungal immune escape, we wish to emphasize that our claim is less dramatic and more specific. Fungal pathogens are many and varied, each having its own immune evasion strategy. *Candida albicans* evolved many immune evasion mechanisms, some known and some yet to be discovered. We think that the interaction with TIGIT is indeed an important immune evasion mechanism, and it is definitely novel since the fungus hijacks immune checkpoint receptors for its needs, but we do not claim it is the central avenue for immune evasion.

The additional figures clearly improve the manuscript. Changes accepted as they are.

Issues in details: 1. Als has several family members. Does each Als proteins bind to the TIGIT receptors with similar affinity and is this interaction specific for human NK cells. What is the common binding motive.

The issues reviewer #2 raises here are indeed important to the understanding of the biology behind the phenomenon we describe in our manuscript.

1. The Als protein family indeed contains many members, each having a different functional role and different ligands. To evaluate the binding affinities of each TIGIT-Als protein interaction we cloned them, produced recombinant versions of these proteins, and used both FACS and MST to test whether they bind to TIGIT and with what affinity (fig. 3c-d and the new fig. 4). Using MST we observed different binding affinities: with Als6 being the strongest TIGIT binder and Als9 being the weakest (new figure 4). This is also in line with the BW experiment (fig. 3a) that shows a more pronounced reduction in TIGIT activation for the Als6-deleted candida strain relative to the Als9-deleted strain (even though they both activate TIGIT significantly less than the WT strain).

2. This described interaction is not specific to human NK cells, as a recombinant TIGIT protein from mouse was also able to bind to *C. albicans* cells (fig. 5c-d), and deletion of Als6, Als7 or Als9 was able to abrogate the effects of TIGIT blockade during in vivo infections of mice (fig. 6).

3. Als proteins are usually composed of several domains. When cloning the Als-Ig fusion proteins used in figures 3+4 we only used the N-terminal domain. This was due to technical reasons (additional domains present in Als proteins have high potential for aggregation) and also due to functional reasons; the N-terminal domain is known to be the main ligand-binding domain for Als proteins and contains the motif which is responsible for binding all of the known Als ligands. This was determined using both functional data and structural data that is available for several Als proteins including Als9-2 (Salgado, P. S., Yan, R., Rowan, F., *et al.* (2011) 'Expression, crystallization and preliminary X-ray data analysis of NT-Als9-2, a fungal adhesin from *Candida albicans*', *Acta Crystallographica Section F: Structural Biology and Crystallization Communications*), (Salgado, P. S., Yan, R., Taylor, J. D., *et al.* (2011) 'Structural basis for the broad specificity to host-cell ligands by the pathogenic fungus *Candida albicans*', *Proceedings of the National Academy of Sciences*).

As reviewer #2 raises an interesting and important point that was not emphasized properly in the text, we edited the relevant parts in the text to further elaborate on it (see pages 11-12 and 16).

Changes accepted as they are.

2. Do all members of the ALS family proteins binds to TIGIT.

In order to check which Als family members bind TIGIT we examined deletion mutants for each and every Als gene in *C. albicans* using a Bw assay and found that only Als6, Als7 and Als9 activate TIGIT (fig. 3a). We also cloned recombinant versions of these proteins to validate and quantify this finding, which led us to also determine that only one allele of Als9, Als9-2, but not Als9-1, is a TIGIT ligand.

The deletion experiments shown in Figure 3a indicate which ALS is responsible or crucial for TIGIT activation. However, these data only indirectly exclude binding of ALS proteins to TIGIT. For example, when als6 mutant is used, no TIGIT activation (due to no binding) by ALS6 is seen. However, in als7 and als9 mutants the ALS6 protein is still present and most likely binds to TIGIT, but other events (ALS7 or ALS9 binding) prevail over the ALS6 effect (same for ALS7 and ALS9). Therefore, the individual binding of each of the ALS proteins can not be concluded from this data. It would be nice to see binding experiments as shown in Figure 4 for other members of the ALS family, but I understand that this would require time. At least, the authors should comment on this limitation.

3. Also T cells express TIGIT receptors, what is the effect of the candida protein's on these central type of immune human cells.

We thank reviewer #2 for this comment. T cells indeed play a central role in anti-fungal immunity. To test whether *Candida albicans* uses TIGIT activation to inhibit T cells in addition to NK cells we performed the experiment described in new figures 2c+d. In short, we isolated primary CD4+TIGIT+ T cells from the blood of human donors, activated them, and checked whether *Candida albicans* cells inhibit them, and whether this inhibition can be abrogated using an anti-TIGIT antibody. We saw that TIGIT blockade could indeed increase the activation levels of T cells (new figure 2d). In addition, we also examined the role of T cells in our in-vivo invasive candidiasis model (figures 5a+b) and observed a significant role for T cells as well. These new results lead us to suggest that the TIGIT-mediated immune evasion of *Candida albicans* works via inhibition of not only NK cells, but also T cells. We now also discuss this in the results and discussion sections (see pages 8, 12-13 and 15-16).

Changes accepted as they are.

4. Does the mAB used here block binding of the recombinant TIGIT to candida, also block the binding of each Als protein to TIGIT in the same manner. Does the antibody show the same inhibitory effect for all Als proteins.

The ability of the anti-TIGIT antibody we use here to block TIGIT was indeed only indirectly shown (in figure 3b for example). In order to directly show the ability of the antibody to block the interaction with the Als proteins and to test whether it has the same effect on all TIGIT-Als interactions, we repeated the experiment depicted in figure 3c+3d (staining of TIGIT-expressing BW cells with recombinant Als-Ig fusion proteins), in the presence of the anti-TIGIT antibody. As can be seen in new figure 3e, the antibody indeed blocks these interactions.

Regarding the strength of the inhibitory effect, figure 3e show that the anti-TIGIT antibody has more or less the same effect on all NT-Als-Ig fusion proteins. Figure 3b shows that this effect is also similar functionally, as deletions of either Als protein completely abolished the effects of TIGIT blockade on the *Candida* cells' immune evasion abilities.

5. The authors use fusion proteins in most experiments, they should include additional assays in which native untagged proteins are used, e.g. for immunoprecipitation.

Many experiments in the manuscript indeed include fusion proteins that include an Ig-tag to identify and quantify them. This tag is also critical in the generation of these proteins as it is used in the protein G affinity column, leading to very high purity (near 100%). The use of untagged proteins is indeed more favorable and can prevent some experimental artifacts. We wished to perform such experiments, but unfortunately, to the best of our knowledge no antibodies against Als6, Als7 or Als9 are available commercially or described in the literature. We also attempted to manufacture these proteins using other tags, but were unsuccessful. With that in mind, we tried to control for the

presence of the Ig tag using several methods: 1. We used another reagent, the Candida Als deletion mutants, to examine the interaction between Als proteins and TIGIT independently of the tagged Als proteins (fig. 3a+b and fig, 6).

2. We performed MST experiments using only TIGIT and Als recombinant proteins to evaluate the Als-TIGIT interaction directly and exclude possible artifacts originating from the use of antibodies or from additional variables that might be present on human or candida cells (please see new figure 4).

3. We used a control fusion protein containing an identical Ig tag, produced in the exact same manner as a negative control in all experiments that included the NT-Als-Ig fusion proteins.

4. In addition to our extensive experience with Ig-tagged proteins (We have used them in many published projects over many years, for example (Mandelboim, O. *et al.* (2001) 'Recognition of haemagglutinins on virus-infected cells by NKp46 activates lysis by human NK cells.', *Nature*), (Stanietsky, N. *et al.* (2009) 'The interaction of TIGIT with PVR and PVRL2 inhibits human NK cell cytotoxicity.', *Proceedings of the National Academy of Sciences of the United States of America*), (Vitenshtein, A. *et al.* (2016) 'NK Cell Recognition of *Candida glabrata* through Binding of NKp46 and NCR1 to Fungal Ligands Epa1, Epa6, and Epa7', *Cell Host and Microbe*), (Reches, A. *et al.* (2020) 'Nectin4 is a novel TIGIT ligand which combines checkpoint inhibition and tumor specificity', *Journal for Immunotherapy of Cancer*), we also went over the literature (as this tag is also widely used in many other labs) and did not find evidence for possible artifacts or effects on the tagged proteins that could be relevant to our study.

Reply accepted, no action required.

6. Is this a uniform mechanism of fungal Als function? Do other fungi, like *Aspergillus fumigatus* use the same type of immune escape. Do *Aspergillus fumigatus* Als proteins, bind to the same human and mouse TIGIT ligands and do they have the same biological effects.

Reviewer #2 raises an interesting question, concerning both the biological mechanism of the interaction we describe and the biological scope of this finding.

In order to answer this question, we performed another experiment described in new fig. 1d and tested whether TIGIT-Ig binds to non-albicans candida species. To further emphasize this point, we now used pathogenic fungi that are not candida; *Cryptococcus neoformans* and *Cryptococcus gattii*. These two species were not recognized by TIGIT-Ig (new fig 1d). This strengthens our conclusion that this immune evasion mechanism is species-specific and not all pathogenic fungi has it. This finding is also in line with our recognition of Als proteins as the TIGIT ligands, as these proteins are very species-specific and are not present in non-candida pathogenic fungi (Hoyer, L. L. and Cota, E. (2016) 'Candida albicans agglutinin-like sequence (Als) family vignettes: A review of als protein structure and function', *Frontiers in Microbiology*). Moreover, even within the Candida family not all species encode Als genes, and only the ones that are known to contain Als (*Candida albicans* and *Candida parapsilosis* (Oh, S.-H. *et al.* (2019) 'Agglutinin-Like Sequence (ALS) Genes in the *Candida parapsilosis* Species Complex: Blurring the Boundaries Between Gene Families That Encode Cell-Wall Proteins', *Frontiers in Microbiology*) were able to bind TIGIT. We now also discuss this in the discussion section (see page 17).

The experiments performed with selected non-Candida fungi do of course not exclude that other fungi like *Aspergillus fumigatus* use this type of immune escape, but incorporation of *Cryptococcus* improves the manuscript.

7. What is the signaling pathway initiated by Als proteins and what is the cell response. Is the effect the same in NK cells and in T lymphocytes? The immune dampening effect need to be evaluated in greater detail.

The intracellular signaling cascade following TIGIT activation is indeed an interesting point that we did not touch in the original version of the manuscript. The signaling cascade initiated by the binding of TIGIT to its endogenous ligands was extensively examined in the past, by both our lab and others (Stanietsky, N. *et al.* (2009) 'The interaction of TIGIT with PVR and PVRL2 inhibits human NK cell cytotoxicity.', *Proceedings of the National Academy of Sciences of the United States of America*), (Liu, S. *et al.* (2013) 'Recruitment of Grb2 and SHIP1 by the ITT-like motif of TIGIT suppresses granule polarization and cytotoxicity

of NK cells', *Cell Death and Differentiation*). This signaling cascade was mostly studied in NK cells, and while it is less studied in T cells, it is hypothesized to be similar in its initial stages. This signaling cascade includes phosphorylation of critical tyrosine residues in the cytoplasmic tail of the protein, recruitment of adapter proteins and intracellular phosphatases, and dampening of signals produced by various kinases.

In order to examine whether the signaling cascade initiated by Candida activation of TIGIT is similar to the one described for the endogenous ligands we used TIGIT-expressing YTS cells mutated in the ITIM domain which is critical for the initiation of the signaling cascade. We used two mutants: one with a point mutation in the ITIM's tyrosine residue (YTS TIGIT Y231A), and one with a stop codon interrupting the ITIM and the final 13 amino acids following it (YTS TIGIT Y231Stop). This model was used in the past by our lab and others to characterize the signaling events responsible for the inhibitory effects of TIGIT (Stanietsky, N. *et al.* (2009) 'The interaction of TIGIT with PVR and PVRL2 inhibits human NK cell cytotoxicity.', *Proceedings of the National Academy of Sciences of the United States of America*), (Liu, S. *et al.* (2013) 'Recruitment of Grb2 and SHIP1 by the ITT-like motif of TIGIT suppresses granule polarization and cytotoxicity of NK cells', *Cell Death and Differentiation*). As can be seen in new figure 2e, any interference with the ITIM motif completely abrogated the Candida-mediated TIGIT inhibition. We now also discuss this in the results and the discussion sections (see pages 8-9 and 17).

Changes accepted as they are.

8. Also in this regard the Als-TIGIT Interaction may be relevant for mediated NK cell activity. But the effect on T cells needs to be included. In this regard, if the concept of Als-TIGIT interaction is really important than the effect on the host immune response mediated by NK and by T cells needs to be explored

We completely agree with reviewer #2. As mentioned in point #3, we have added both in vitro and in vivo data regarding the effects of the Candida-TIGIT interaction (please see new figures 2c-d and 5a-b).

Changes accepted as they are.

9. Figure 1. One problem in this figure is presented in panels 1b, 1c, and 1d by the presentation of relative values. What are the real binding data. Like in 1c, a value two times about control may not be a dramatic high affinity binding.

We thank reviewer #2 for this important comment. We also agree that presentation of raw data together with relative values is important in order to report the data in a way that combines and maximizes both transparency and comprehensibility. To that end, we included a histogram showing representative raw data with most experiments (for example Fig. 1a, 3c and 5c).

To provide the reviewer with the requested values we bring below graphs showing the raw data corresponding to figures 1b, 1c and 1d. One of the main reasons we choose to show quantifications of multiple FACS and ELISA experiments using data relative to a constant control is due to high inter-experimental changes occurring due to various technical issues such as using different machines, different settings, different experimenters and so on. This can also be observed in the raw data presented below. To present the data accurately we changed the Y-axis from linear to logarithmic in the following graphs (in the FACS experiments). In addition, one specific data point in fig. 1b is significantly different from the rest, so we present the data with and without it (fig. 1b1 vs 1b2).

Reply accepted.

10. For panels 1e and 1f. Whether controls include like using the mAb to block the interaction. How is the effect on binding to the various als knock out mutants. In particular given the redundancy of the various Als proteins this aspect needs to be addressed .

Panels 1e and 1f present an immunofluorescence microscopy experiment in which we stained yeast and hyphae candida cells to understand whether the binding we observed between TIGIT and candida yeast cells is also present in other morphologies of *C. albicans*. Due to that, we only

performed controls necessary for the technical aspect of experiment (such as using a control-Ig protein as a negative control and staining yeast cells as a positive control). We agree with reviewer #2 that additional characterization of this interaction, including the specific effects of the different Als ligands or the blocking antibody we use in experiments in later parts of the manuscript are indeed important. We tried using confocal microscopy to characterize and quantify these interactions, but due to the limitation in quantification inherent in the method, we did not succeed, although a trend was observed. Therefore, we prefer showing more reliable quantitative assays, such as FACS, ELISA, BW and NK cytotoxicity (fig. 3a,b,c,d for the effect of the different Als ligands and fig. 2a, 2b, 2d and 3e for the effects of the antibody). If reviewer #2 still feels that the microscopy experiment presented in fig. 1e-f is missing critical controls, we will repeat the confocal experiments many times to obtain statistical differences.

Reply accepted.

11. Figure 2. What is the meaning of the clinical strains. The clinical strains are an impressive selection. It would be nice to compare the expression levels of the various Als isoforms at least in several of the high and low effective isolates.

The mechanism behind the differential ability of the clinical *C. albicans* isolates to activate TIGIT is indeed an interesting question. We checked the Als mRNA expression levels of the five most activating isolates and the three weakest activating isolates using quantitative real-time PCR (please see figure below). As controls we also measured the Als expression levels in the WT lab strain (SC5314) and its three daughter strains deleted for Als6, Als7 or Als9.

As can be seen in the figure, we did not identify any significant differences. This suggests that another mechanism, most likely a post-transcriptional one, is responsible for the difference in TIGIT activation abilities in the examined strains. Such differences can be on the protein level, for example differences in translation efficiencies, protein degradation rates, protein modifications, protein localization, and so on. Another possible might be that the Als proteins work as a complex and could be regulating the activity and/or expression levels of each other, creating a complex regulatory network between themselves (Hoyer, L. L. and Cota, E. (2016) 'Candida albicans agglutinin-like sequence (Als) family vignettes: A review of als protein structure and function', *Frontiers in Microbiology*). Unfortunately, to the best of our knowledge no antibodies are available that directly bind Als6, Als7, Als9-1 or Als9-2. As such, probing differences on the protein level is highly complicated. In addition, it is likely that each clinical isolate evolved a different mechanism to change its Als protein activity. We now discuss this in the discussion section (see page 17).

Reply accepted.

12. Figure 3. The effect TIGIT activation her all nine als knock out mutant were assessed What is the contribution of each als isoform on expression level and on TIGIT mediated activation. Please show also the raw data and not only the relative effects.

The aforementioned experiment is complex. For example, we realized that TIGIT activation levels are affected by the specific growth conditions of the BW cells, and that the live Candida cells affect the BW cells and their ability to be activated and secrete IL-2, probably due to the sensing of fungal metabolites and the secretion of fungal proteases, etc..

Due to that, we analyzed the experiments' results using three stages.

First, we perform the co-incubation of the Candida cells with either parental BW cells or TIGIT-expressing BW cells. BW cell activation is measured by the secretion of IL-2 (using anti-IL-2 ELISA). Next, we normalize the results from each BW TIGIT x Candida sample to an identical reaction with the same Candida strain in the presence of the BW parental cells. This allows us to better control for the various non-TIGIT related effects and artifacts. Finally, we normalize these results to the measured values of the BW TIGIT x WT Candida in each experiment, to control for inter-experimental variability (which tends to be quite high sometimes). From our experience this experimental scheme provides accurate and better controlled results. Because of the reviewer's comment, we understood

that this process isn't explained well enough in the manuscript, so we edited and improved it (please see the relevant methods sub-section, pages 28-29). In addition, we provide below the raw data obtained from the different stages:

Data analysis stages for the BW activation assay. TIGIT activation was assayed using the murine thymoma cell line BW expressing a chimeric TIGIT-z-chain receptor. The BW cells were co-incubated for 48 hours in the presence or absence of the WT *C. albicans* strain SC5314 or mutant strains deleted for members of the Als protein family. Identical wells containing parental BW cells not expressing TIGIT were also prepared for each experimental condition. TIGIT activation was measured using ELISA for IL-2 secreted by the activated BW cells. Shown are averages of 2-4 independent experiments. The upper figure shows the raw OD values read from the ELISA plates. The middle figure shows these values after normalization to their matched parental BW samples for better control of various internal experiment noise. The lower figure shows the final data after normalization to the WT candida strain in each experiment, in order to better control for noise and technical differences between different experiments.

Reply accepted

13. Figure 4. The effect a control protein (PVR-Ig) is comparable to Als7 protein. Does this reflect a specific interaction and high affinity binding? As negative control and lacking IgG domain needs to be included. In addition, k_{on} and k_{off} values and rates need to be presented. A critical interpretation of this results is that the also Ig fusion proteins show related binding at the human ligand PVR Ig. However, a negative control needs to be presented.

1. Indeed, the MST results for the interaction between TIGIT and both NT-Als7-Ig and PVR-Ig are very similar (40-50nm), with the binding affinity for NT-Als6-Ig even stronger (6nm) and the affinity for NT-Als9-2-Ig weaker (455nm). All these interactions are within an order of magnitude lower than our positive control; the well characterized interaction between TIGIT and its endogenous ligand PVR. As such, we conclude that all these interactions are biologically relevant. Additional support for this conclusion is the fact that the binding affinities between the Als-Ig fusion proteins and TIGIT are similar to those observed in the BW experiment (fig. 3a). Deletion of Als6 leads to the strongest effect on TIGIT activation and deletion of Als9 leads to the weakest. We have also edited the relevant part in the text to better clarify and emphasize these points (see page 12).

2. We agree that a negative control should be included. A negative control was of course included in the original experiment but not shown. This is fixed now, and the new figure 4 includes it.

3. Potential artifacts originating from the IgG Fc domain were controlled by using a negative control protein that includes an Ig tag, and now appears in the new figure 4 and the text (see page 12). In addition, we feel confident that it does not significantly affect the interaction with TIGIT-Ig as we have performed several similar MST experiments with various ligands of TIGIT fused to this tag in different published projects over the years (for example in (Reches, A. *et al.* (2020) 'Nectin4 is a novel TIGIT ligand which combines checkpoint inhibition and tumor specificity', *Journal for Immunotherapy of Cancer*).

4. Unfortunately, K_{on} and K_{off} values are not possible to get when using MST (this is a known limitation of the instrument).

Reply and changes accepted as they are.

14. Figure 5. The effect Figure 5a and 5b: Figure 5d again please present the experimental data without the calculation of the value relative to control. What is the effect in T cell depleted animals.

1. New T cell data was added to new figures 5a and 5b, showing that T cells are also involved the control of invasive candidiasis. In accordance with these results we added further discussion of this subject to various sections of the manuscript (see pages 8-9 and 15-16).

2. We agree with the reviewer that it is important to take into account both the raw data and the relative one to properly assess the reported effect. As mentioned in a previous comment, we initially added the representative histogram (fig. 5c), in addition to the quantification of the relative results (fig. 5d). We think that the combination of the two data formats provides the best balance between data transparency and comprehensibility.

In addition, we show below a quantification of the raw data without normalization to the control-Ig. Please notice that the Y-axis was converted into log scale in order to better present the data. This is due to high variability in the signal strength between the experiments (due to using different FACS machines at different times using slightly different technical settings) and can be an example for our reasoning behind presenting relative data in addition to the histogram presenting the raw data.

Reply accepted.

15. Figure 6. The effect of the various *C. albicans* AlsI knock out mutant in the in vivo infection model need to be evaluated and discussed first. Then the role of mAB can be addressed first. What is the reason on selecting got the three mutants, als6, als7 and als 9. Additional control like revertant strains for each knock out mutant studied need to be presented.

1. We re-ordered the figure and its corresponding parts in the manuscript accordingly, as suggested (see pages 13-14).

2. We chose to select the three Als mutants, *als6Δ/Δ*, *als7Δ/Δ* and *als9Δ/Δ* for further in-vivo evaluation as the experiments depicted in figures 3 and 4 indicate that they are the major ligands of TIGIT present on *C. albicans*. We thought that in vivo experiments comparing the effects of TIGIT blockade on the mutants and WT strains will further validate their role as TIGIT ligands under physiological conditions. Another reason behind their inclusion was that they also serve as a control for the possible side effects of TIGIT blockade during invasive candidiasis. It is possible that TIGIT blockade can unspecifically disinhibit the immune response, as seen as a common complication of immune checkpoint inhibition in human patients. If that were to happen in our model, we would have expected to see the same effect during TIGIT blockade in all treated mice, independent of the candida strain infecting them. On the other hand, if the effects of the anti-TIGIT antibody are specific to the Als immune evasion phenomenon, and are not due to the potential general immune-disinhibitory effect of checkpoint inhibitors, the usage of the ligand-missing mutants would abrogate this effect. We edited the text to better describe these points (see pages 13-15).

3. Adding revertants is indeed a nice suggestion but it require a very long time to complete these experiments (in our opinion more than a year, as we have to generate them and then test them using the various in vitro and in vivo assays). As we are already working on this project for around five years, we hope that the reviewer will agree to the current manuscript being published without the revertants. We will work on them in the future related projects.

Changes and comments accepted as they are.

16. Language: The text needs revision in language and also in content. Some of the statements sound rather dramatic and in the end are not absolutely right. E.g. The point 'that *Candida albicans* is ... one of the deadliest causes of bloodstream infections' sounds very dramatic but is not complement right (page 2, abstract first sentence). Do the authors really want to say that other pathogenic bacteria like *Streptococcus pneumoniae*, *S pyogenes*, *Staphylococcus aureus* are not relevant. In particular not to mention the current COVID-19 situation which gives another example on severay of blood stream infections.

We thank the reviewer for this critical point as we always wish to report our data in an accurate and leveled fashion. As such, we went over the text and modified the relevant statements (see page 2 and additional statements throughout the paper).

Regarding the specific point the reviewer raised, we referred to the data presented in the introduction showing that *C. albicans* infection is extremely common, being the 3rd to 4th most common cause of bloodstream infection in hospitalized patients in many countries (and being responsible for 18-22% of such infections in the US). It is also extremely deadly, as 40-60% of all infected patients die (Pappas, P. G. *et al.* (2018) 'Invasive candidiasis', *Nature Reviews Disease Primers*). We did not, by any means, try to imply that other pathogens, whether they be bacteria such as *Staphylococci* or *Streptococci*, viruses such as SARS-COV-2, or other fungi or even parasites, are less relevant.

Unfortunately for patients, all of these pathogens are still equally deadly and important and lead to high morbidity and mortality. The case of COVID-19 which the reviewer brought up is of specific interest, not only due to the global pandemic, but also due to the fact that like most intensive care admitted patients (and especially those treated with steroids), individuals suffering from COVID-19 are at a very high risk of fungal bloodstream infections which takes the lives of many of them. In order to prevent further misunderstandings regarding this statement we amended and softened it, and went over the text to amend similar statements.

Changes accepted as they are.

Reviewer #1 (Remarks to the Author):

Charpak-Amikam et al report on the identification of the TIGIT receptor as the NK cell receptor binding to *Candida albicans* yeast and hyphae. Via this receptor, NK cells recognize the *C. albicans* cell wall-bound adhesion proteins Als6, Als7 and Als9. The ligands binding to this receptor lead to inhibition NK cell function, and in vivo studies in mice infected with *C. albicans* demonstrate higher mortality. As fungal infections are still a major cause of morbidity and mortality in different patient populations, the present data are of high interest for a better understanding of the host-fungal interaction and may provide the rationale for new therapeutic strategies.

Although the data are well presented and quite convincing, I would like to make several comments.

We thank reviewer #1 for the thoughtful and interesting review. Following reviewer 1 and 2 comments 6 new figures were added (fig. 2c,d,e, 3e and S1c-d) and 5 more were modified and improved (fig. 1d, 3b, 4 and 5a-b). We have amended the manuscript in accordance with the following comments:

1. The authors state that *C. albicans* binds NK cell TIGIT and inhibits NK cell activity in vitro, as demonstrated in Figure 2b. This was evaluated using 1) YTS cells, which naturally do not express TIGIT (YTS Eco), and in which blocking TIGIT does not have any impact 2) YTS-TIGIT cells in which the authors demonstrate that *C. albicans* proteins bind to the TIGIT receptor, resulting in the inhibition of the cellular antifungal activity. In contrast to YTS Eco, the addition of α -TIGIT antibody increased (restored) the antifungal activity of YTS-TIGIT cells. Nevertheless, it remains unclear why the anti-*Candida* activity is lower in YTS Eco cells not expressing TIGIT compared to that of YTS-TIGIT cells after addition of an α -TIGIT antibody. Does the α -TIGIT antibody exhibit activating properties? Does the TIGIT receptor have activating properties and does the binding of the α -TIGIT antibody boost the antifungal activity of the transduced (YTS-TIGIT) cells? From the data presented in Figure 2a one cannot conclude that TIGIT is “functional and inhibitory for NK cells” (page 7, line 151). Authors, please address these comments/questions.

Reviewer #1 raises two important issues:

1. Regarding the difference in anti-candidal activity between the YTS Eco and the YTS TIGIT cells. We also expected that the addition of anti-TIGIT to the YTS TIGIT cells would lead to similar anti-fungal activity as observed in YTS Eco cells. We think that the differences are due to clonality of the different YTS cell lines. After the transgene encoding for TIGIT was transduced into the parental YTS Eco cell line, a single TIGIT-positive clone was chosen and further propagated to establish the YTS TIGIT line. As a result, it might behave slightly different as compared to the parental YTS Eco cells and express differently other receptors (such as 2B4, see figure below) that are important for YTS killing.

Due to this reason, in fig. 2a we don't compare directly between the two lines, but only within each of the lines separately, and compare the isotype control to the anti-TIGIT antibody. We now discuss this point in more details in the manuscript (See page 7).

2. Regarding the reviewer's comment that "*From the data presented in Figure 2a one cannot conclude that TIGIT is "functional and inhibitory for NK cells" (page 7, line 151)*".

We completely agree. We based our conclusion on figure 2b which shows the effect of TIGIT on the anti-fungal activity of primary NK cells obtained from several human donors. Following these comments, we have edited the text to emphasize and clarify these points (Pages 7-8).

Changes accepted - no further comment.

2. The y-axes in Figure 3b show different ranges, which means that the antifungal activity of the different mutants is different. Authors, please adjust the y-axes to allow a better comparison of the mutants and explain these differences.

We thank the reviewer for this comment. The relevant axes were adjusted accordingly. It is now easier to observe that there isn't a major difference between the antifungal activities against the various mutants. The minor differences that are still present are due to the fact that each experiment shown in figure 3b was performed using primary NK cells isolated from different individuals. As the reviewer knows each primary NK cell line expresses NK activating and inhibitory receptors differently and thus have a different basic activity level, not only against fungi but also against various other tumor targets. Importantly however, when each NK line is compared to itself, it can be seen that they all respond to TIGIT blockade unless the *Candida* strain used is missing one of the TIGIT ligands.

Changes accepted. The adjustment of the axes makes it easier to compare the data and I appreciate that the authors followed my suggestion.

3. The authors report that NK cells express TIGIT, which binds Als6/7/9 of *C. albicans* as ligands. The binding of these proteins (Als6/7/9) leads to the inhibition of NK cell activity, both in vitro and in vivo. α -TIGIT antibody counteracts this binding and therefore also NK cell inhibition by Als6/7/9, resulting in increased antifungal activity of NK cells. Mutant *C. albicans* strains lacking als 6, 7, or 9 (als6/7/9 Δ/Δ) do not inhibit NK cell activity, which demonstrates the importance of these fungal proteins in decreasing NK cell activity. In Figure 6c, it remains unclear why als6 Δ/Δ *C. albicans* and als7 Δ/Δ *C. albicans* (still expressing Als7 and 9 or Als6 and 9, respectively) do not inhibit NK cells, whereas als9 Δ/Δ (still expressing Als6 and 7) exhibits NK cell inhibition in vivo (Figure 6c), resulting in high mortality in *C. albicans* infected mice. Notably, the mortality is not significantly different from that of the wild type *C. albicans*. In addition, what is the impact on antifungal activity and mortality if combining different mutants (e.g., als6 Δ/Δ and als7 Δ/Δ or als6 Δ/Δ and als9 Δ/Δ etc)?

1. We have also wondered about the different in virulence observed between the mutant

strains for the Als6, Als7, and Als9 proteins. The Als proteins are well-known virulence factors playing an important role in adhesion. As each member of the Als family is expressed in different levels at different infection sites and times, it is hypothesized that each one fulfills different functions during *C. albicans* invasion. As such, it was previously demonstrated that each Als member can have several functions and several ligands (Hoyer, L. L. and Cota, E. (2016) 'Candida albicans agglutinin-like sequence (Als) family vignettes: A review of als protein structure and function', *Frontiers in Microbiology*), (Klotz, S. A. et al. (2004) 'Degenerate Peptide Recognition by Candida albicans Adhesins Als5p and Als1p', *Infection and Immunity*). Thus, while both Als6, Als7, and Als9 bind TIGIT, each one of them also bind different other proteins. As a result, deletion of each Als protein leads to a different effect on virulence.

Under the conditions used our experiment (CFU injected per mouse, the specific mouse strain, and for 6a [6b after this revision] the specific organ examined and time of examination, etc.) the role of Als6 and Als7 is more pronounced, as their deletion cripples the virulence of *C. albicans*. Deletion of Als9 did not lead to a significant change in virulence. This is probably due to the numerous effects of Als9 deletion; it prevents TIGIT activation, but also changes the adhesion properties of the cells, changing their organ tropism in the mouse and the clinical disease the fungus causes. This leads to a complex phenotype, in which the lack of ability to activate TIGIT is probably balanced with the different adhesion and tissue tropism properties of the Als9 deletion mutant, resulting in an equal virulence to the WT strain. Similar effects have been shown for other Als proteins, that when deleted can in some conditions paradoxically increase adhesion (Zhao, X., Oh, S. H. and Hoyer, L. L. (2007) 'Deletion of ALS5, ALS6 or ALS7 increases adhesion of Candida albicans to human vascular endothelial and buccal epithelial cells', *Medical Mycology*).

Therefore, to differentiate between the various functions of the Als proteins and their effect on TIGIT, we compared between mock treatment and TIGIT blockade of each of deletion mutant separately (in 6a and 6b [6b and 6c after revision]). This enabled us to observe that deletion of each of the Als proteins led to a loss of effect of the anti-TIGIT treatment, even though the background effect of the deletion is different for each Als gene deleted.

As we think this point that the reviewer brought up is indeed important for understanding of our results and was not explained clearly enough in the original text. We now better explain it in the revised manuscript (see pages 14-15).

2. The idea of combining different Als deletion mutants for the in vivo infections is indeed an interesting one. We also considered performing such an experiment but have decided against it due to several reasons. First, it can be seen that deletion of each of the single Als proteins that act as a TIGIT ligand is enough to completely abrogate the effect of TIGIT blockade. Coinfection of the mice with a combination of strains, or creating new strains deleted for several Als proteins, will most likely still maintain the observed complete elimination of this effect,

and so will not fundamentally change our understanding of the reported phenomenon. Therefore, it is considered as unethical and will not be approved by our ethical committee.

Reviewer #1's question raises another interesting issue; why a deletion of each single Als member is enough to obtain a full effect? We think this is an important and interesting issue, and we now discuss it in our discussion section. Our main hypothesis, which was also raised by others (Hoyer, L. L. and Cota, E. (2016) 'Candida albicans agglutinin-like sequence (Als) family vignettes: A review of als protein structure and function', *Frontiers in Microbiology*), is that the Als proteins work as a complex. While all three members can independently bind TIGIT (as shown in figures 3+4), they are non-redundant and each one plays a critical role. Another option that was also suggested previously (Hoyer, L. L. and Cota, E. (2016) 'Candida albicans agglutinin-like sequence (Als) family vignettes: A review of als protein structure and function', *Frontiers in Microbiology*) is that Als proteins regulate the expression levels of other members of the family, and the deletion of one lead to significant changes in expression of the others. We now discuss this in the discussion section (see page 16).

4. The authors state in the discussion section (page 14, line 297-298) that "deletion of just one of them *Als6, 7 or 9+ is enough to completely abolish their effect on TIGIT" (page 14, line 297-298). However, data from Figure 6b and 6c show that deletion of als9 does not protect the mice from *C. albicans* infection, since the survival curve of the mice is similar of that of WT *C. albicans*, and therefore, this finding does not support this statement.

This important point Reviewer #1 raises is linked to our answer for the previous point. What we meant in the abovementioned sentence is that the deletion of each Als member (Als6, 7 or 9) is enough to abrogate the effect of TIGIT blockade, but as each Als member also affects virulence, their deletion leads to other effects unrelated to TIGIT activation. Due to these other effects, deletion of Als6 and Als7 leads to a more prominent reduction in virulence relative to Als9 deletion. This is why we compare each strain to itself with and without TIGIT blockade in figure 6. As mentioned in the previous point, this point was not explained well. We now better explain it in the results and discussion sections (see page 14-16).

Points 3 & 4:

I agree with the author's explanation why not to combine different Als deletion mutants and Als proteins working as a complex. However, I still hesitate with the role of Als9. While the title of the according paragraph is "The Als-mediated immune evasion mechanism can be targeted using immunotherapy in mice", the data demonstrated by the authors do not support Als9 as a target. The mouse experiment show that targeting Als9 has no effect on the virulence of the fungus and the survival of the animals. Although the deletion of als9 had an effect in the cytotoxicity experiments in vitro (Fig 3b), this effect was not seen in vivo. In line 302-303 the authors state "... under our experimental conditions *als9Δ/Δ* appeared to not be significantly more pathogenic relative to the WT strain (Fig.6A)". In addition, the authors state that "The *als9Δ/Δ* strain did lead to significant mortality and kidney fungal burden in the

infected mice, and surprisingly did not show a significantly reduced pathogenicity relative to the WT strain.” (line 315ff). These facts suggest that blocking Als9 in vivo does not make a difference on the outcome of the infection. Similarly, “... the net effect [of] Als9 deletion, which consists of its pro- and anti-adhesion effects and its TIGIT activating abilities, is neutral.” (line 322). Therefore, I still wonder why Als9 should be a good target for immunotherapy.

We completely agree with the point made by the reviewer, which we think highlights a misunderstanding deserving a better explanation by us. As our data shows that a Als9 deletion does not interfere with *C. albicans* pathogenicity (at least under the experimental conditions we used) it is indeed not a good target for blocking. The blocking target we refer to in the paper is only TIGIT, which we show has an effect when blocked (Fig. 6b+c). Our conclusions regarding the Als proteins, including Als9, is that they are involved and perhaps responsible for this phenotype and for the effects of the TIGIT blockade. We do not discuss blocking of the fungal TIGIT ligands, although it probably could be a relevant therapeutic avenue once blocking antibodies against them will be developed. In such a hypothetical case, we agree with the reviewer that Als9 would probably be a less promising target.

To better clarify this point, we changed the title of the relevant results section, the conclusion derived from it at the end of the discussion section, and also re-examined the results & discussion sections to make sure that we clarify that only TIGIT blockade is discussed.

Reviewer #2 (Remarks to the Author): The manuscript by Yoav Charpak-Amikam and colleagues aims to describe a new fungal immune evasion step of the human pathogenic yeast *Candida albicans*. The authors approach how *Candida albicans* can inactivate and dampen the immune attack of human natural killer cells. The group focuses on the immune inhibitory receptor expressed on the surface of human NK cells, TIGIT, and they use a screening approach to identify the fungal ligands. Upon screening of a fungal deletion library, the authors identify candida Als family proteins as ligands for both human and mouse TIGIT receptors. In addition, they used an in vivo infection model where they approach the role of several Als isoforms, and furthermore they use a monoclonal antibody to block Als protein function. This concept is of interest and may ultimately show a new avenue of fungal and in particular candida immune evasion. However, in the present form, additional work is required to clearly and unambiguously confirm the concept presented by the authors. Numerous additional controls experiments are necessary and need to be presented in order to provide a clear evidence that candida Als proteins interact with the immune inhibitory receptor TIGIT expressed on human and mouse NK, as well as T cells. In addition further proof is necessary to conclude that the TIGIT -Als interaction is the central avenue of fungal immune escape.

We thank reviewer #2 for the interesting and constructive review. Following reviewer 1 and 2 comments 6 new figures were added (fig. 2c,d,e, 3e and S1c-d) and 5 more were modified and improved (fig. 1d, 3b, 4 and 5a-b). We have amended the manuscript in accordance with the following comments.

In addition, regarding the reviewer’s last sentence dealing with the TIGIT-Als interaction as the central avenue of fungal immune escape, we wish to emphasize that our claim is less dramatic and more specific. Fungal pathogens are many and varied, each having its own immune evasion strategy.

Candida albicans evolved many immune evasion mechanisms, some known and some yet to be discovered. We think that the interaction with TIGIT is indeed an important immune evasion mechanism, and it is definitely novel since the fungus hijacks immune checkpoint receptors for its needs, but we do not claim it is the central avenue for immune evasion.

The additional figures clearly improve the manuscript. Changes accepted as they are.

Issues in details: 1. Als has several family members. Does each Als protein bind to the TIGIT receptors with similar affinity and is this interaction specific for human NK cells. What is the common binding motive.

The issues reviewer #2 raises here are indeed important to the understanding of the biology behind the phenomenon we describe in our manuscript.

1. The Als protein family indeed contains many members, each having a different functional role and different ligands. To evaluate the binding affinities of each TIGIT-Als protein interaction we cloned them, produced recombinant versions of these proteins, and used both FACS and MST to test whether they bind to TIGIT and with what affinity (fig. 3c-d and the new fig. 4). Using MST we observed different binding affinities: with Als6 being the strongest TIGIT binder and Als9 being the weakest (new figure 4). This is also in line with the BW experiment (fig. 3a) that shows a more pronounced reduction in TIGIT activation for the Als6-deleted *Candida* strain relative to the Als9-deleted strain (even though they both activate TIGIT significantly less than the WT strain).

2. This described interaction is not specific to human NK cells, as a recombinant TIGIT protein from mouse was also able to bind to *C. albicans* cells (fig. 5c-d), and deletion of Als6, Als7 or Als9 was able to abrogate the effects of TIGIT blockade during *in vivo* infections of mice (fig. 6).

3. Als proteins are usually composed of several domains. When cloning the Als-Ig fusion proteins used in figures 3+4 we only used the N-terminal domain. This was due to technical reasons (additional domains present in Als proteins have high potential for aggregation) and also due to functional reasons; the N-terminal domain is known to be the main ligand-binding domain for Als proteins and contains the motif which is responsible for binding all of the known Als ligands. This was determined using both functional data and structural data that is available for several Als proteins including Als9-2 (Salgado, P. S., Yan, R., Rowan, F., *et al.* (2011) 'Expression, crystallization and preliminary X-ray data analysis of NT-Als9-2, a fungal adhesin from *Candida albicans*', *Acta Crystallographica Section F: Structural Biology and Crystallization Communications*), (Salgado, P. S., Yan, R., Taylor, J. D., *et al.* (2011) 'Structural basis for the broad specificity to host-cell ligands by the pathogenic fungus *Candida albicans*', *Proceedings of the National Academy of Sciences*).

As reviewer #2 raises an interesting and important point that was not emphasized properly in the text, we edited the relevant parts in the text to further elaborate on it (see pages 11-12 and 16).

Changes accepted as they are.

2. Do all members of the ALS family proteins binds to TIGIT.

In order to check which Als family members bind TIGIT we examined deletion mutants for each and every Als gene in *C. albicans* using a Bw assay and found that only Als6, Als7 and Als9 activate TIGIT (fig. 3a). We also cloned recombinant versions of these proteins to validate and quantify this finding, which led us to also determine that only one allele of Als9, Als9-2, but not Als9-1, is a TIGIT ligand.

The deletion experiments shown in Figure 3a indicate which ALS is responsible or crucial for TIGIT activation. However, these data only indirectly exclude binding of ALS proteins to TIGIT. For example, when als6 mutant is used, no TIGIT activation (due to no binding) by ALS6 is seen. However, in als7 and als9 mutants the ALS6 protein is still present and most likely binds to TIGIT, but other events (ALS7 or ALS9 binding) prevail over the ALS6 effect (same for ALS7 and ALS9). Therefore, the individual binding of each of the ALS proteins can not be concluded from this data. It would be nice to see binding experiments as shown in Figure 4 for other members of the ALS family, but I understand that this would require time. At least, the authors should comment on this limitation.

We agree with the reviewer on this issue, as it does highlight a limitation in our study. It is completely possible that additional proteins, including Als1-5, play a role in TIGIT activation and perhaps even bind it. If that is the case, our data only show that this is redundant (as their deletion alone did not reduce TIGIT activation capacity) and Als6/7/9-dependent (as the deletion of Als6/7/9 completely abrogated TIGIT activation in our cytotoxicity and in vivo experiments).

We now highlight and discuss this limitation in our discussion section (pages 16-17).

3. Also T cells express TIGIT receptors, what is the effect of the candida protein's on these central type of immune human cells.

We thank reviewer #2 for this comment. T cells indeed play a central role in anti-fungal immunity. To test whether *Candida albicans* uses TIGIT activation to inhibit T cells in addition to NK cells we performed the experiment described in new figures 2c+d. In short, we isolated primary CD4⁺TIGIT⁺T cells from the blood of human donors, activated them, and checked whether *Candida albicans* cells inhibit them, and whether this inhibition can be abrogated using an anti-TIGIT antibody. We saw that TIGIT blockade could indeed increase the activation levels of T cells (new figure 2d). In addition, we also examined the role of T cells in our in-vivo invasive candidiasis model (figures 5a+b) and observed a significant role for T cells as well. These new results lead us to suggest that the TIGIT-mediated immune evasion of *Candida albicans* works via inhibition of not only NK cells, but also T cells. We now also discuss this in the results and discussion sections (see pages 8, 12-13 and 15-16).

Changes accepted as they are.

4. Does the mAB used here block binding of the recombinant TIGIT to candida, also block the binding of each Als protein to TIGIT in the same manner. Does the antibody show the same inhibitory effect for all Als proteins.

The ability of the anti-TIGIT antibody we use here to block TIGIT was indeed only indirectly shown (in figure 3b for example). In order to directly show the ability of the antibody to block the interaction with the Als proteins and to test whether it has the same effect on all TIGIT-Als interactions, we repeated the experiment depicted in figure 3c+3d (staining of TIGIT-expressing BW cells with recombinant Als-Ig fusion proteins), in the presence of the anti-TIGIT antibody. As can be seen in new figure 3e, the antibody indeed blocks these interactions.

Regarding the strength of the inhibitory effect, figure 3e show that the anti-TIGIT antibody has more or less the same effect on all NT-Als-Ig fusion proteins. Figure 3b shows that this effect is also similar functionally, as deletions of either Als protein completely abolished the effects of TIGIT blockade on the Candida cells' immune evasion abilities.

5. The authors use fusion proteins in most experiments, they should include additional assays in which native untagged proteins are used, e.g. for immunoprecipitation.

Many experiments in the manuscript indeed include fusion proteins that include an Ig-tag to identify and quantify them. This tag is also critical in the generation of these proteins as it is used in the protein G affinity column, leading to very high purity (near 100%). The use of untagged proteins is indeed more favorable and can prevent some experimental artifacts. We wished to perform such experiments, but unfortunately, to the best of our knowledge no antibodies against Als6, Als7 or Als9 are available commercially or described in the literature. We also attempted to manufacture these proteins using other tags, but were unsuccessful. With that in mind, we tried to control for the presence of the Ig tag using several methods: 1. We used another reagent, the Candida Als deletion mutants, to examine the interaction between Als proteins and TIGIT independently of the tagged Als proteins (fig. 3a+b and fig, 6).

2. We performed MST experiments using only TIGIT and Als recombinant proteins to evaluate the Als-TIGIT interaction directly and exclude possible artifacts originating from the use of antibodies or from additional variables that might be present on human or candida cells (please see new figure 4).

3. We used a control fusion protein containing an identical Ig tag, produced in the exact same manner as a negative control in all experiments that included the NT-Als-Ig fusion proteins.

4. In addition to our extensive experience with Ig-tagged proteins (We have used them in many published projects over many years, for example (Mandelboim, O. *et al.* (2001) 'Recognition of haemagglutinins on virus-infected cells by NKp46 activates lysis by human NK cells.', *Nature*), (Stanietsky, N. *et al.* (2009) 'The interaction of TIGIT with PVR and PVRL2 inhibits human NK cell cytotoxicity.', *Proceedings of the National Academy of Sciences of the United States of America*), (Vitenshtein, A. *et al.* (2016) 'NK Cell Recognition of *Candida glabrata* through Binding of NKp46 and NCR1 to Fungal Ligands Epa1, Epa6, and Epa7', *Cell Host and Microbe*), (Reches, A. *et al.* (2020) 'Nectin4 is a novel TIGIT ligand which combines checkpoint inhibition and tumor specificity', *Journal for Immunotherapy of Cancer*), we also went over the literature (as this tag is also widely used in many other labs) and did not find evidence for possible artifacts or effects on the tagged proteins that could be relevant to our study.

Reply accepted, no action required.

6. Is this a uniform mechanism of fungal Als function? Do other fungi, like *Aspergillus fumigatus* use the same type of immune escape. Do *Aspergillus fumigatus* Als proteins, bind to the same human and mouse TIGIT ligands and do they have the same biological effects.

Reviewer #2 raises an interesting question, concerning both the biological mechanism of the interaction we describe and the biological scope of this finding.

In order to answer this question, we performed another experiment described in new fig. 1d and tested whether TIGIT-Ig binds to non-*albicans* *Candida* species. To further emphasize this point, we now used pathogenic fungi that are not *Candida*; *Cryptococcus neoformans* and *Cryptococcus gattii*. These two species were not recognized by TIGIT-Ig (new fig 1d). This strengthens our conclusion that this immune evasion mechanism is species-specific and not all pathogenic fungi has it. This finding is also in line with our recognition of Als proteins as the TIGIT ligands, as these proteins are very species-specific and are not present in non-*Candida* pathogenic fungi (Hoyer, L. L. and Cota, E. (2016) 'Candida albicans agglutinin-like sequence (Als) family vignettes: A review of als protein structure and function', *Frontiers in Microbiology*). Moreover, even within the *Candida* family not all species encode Als genes, and only the ones that are known to contain Als (*Candida albicans* and *Candida parapsilosis* (Oh, S.-H. *et al.* (2019) 'Agglutinin-Like Sequence (ALS) Genes in the *Candida parapsilosis* Species Complex: Blurring the Boundaries Between Gene Families That Encode Cell-Wall Proteins', *Frontiers in Microbiology*) were able to bind TIGIT. We now also discuss this in the discussion section (see page 17).

The experiments performed with selected non-*Candida* fungi do of course not exclude that other fungi like *Aspergillus fumigatus* use this type of immune escape, but incorporation of *Cryptococcus* improves the manuscript.

7. What is the signaling pathway initiated by Als proteins and what is the cell response. Is the effect the same in NK cells and in T lymphocytes? The immune dampening effect need to be evaluated in greater detail.

The intracellular signaling cascade following TIGIT activation is indeed an interesting point that we did not touch in the original version of the manuscript. The signaling cascade initiated by the binding of TIGIT to its endogenous ligands was extensively examined in the past, by both our lab and others (Stanietsky, N. *et al.* (2009) 'The interaction of TIGIT with PVR and PVRL2 inhibits human NK cell cytotoxicity.', *Proceedings of the National Academy of Sciences of the United States of America*), (Liu, S. *et al.* (2013) 'Recruitment of Grb2 and SHIP1 by the ITT-like motif of TIGIT suppresses granule polarization and cytotoxicity of NK cells', *Cell Death and Differentiation*). This signaling cascade was mostly studied in NK cells, and while it is less studied in T cells, it is hypothesized to be similar in its initial stages. This signaling cascade includes phosphorylation of critical tyrosine residues in the cytoplasmic tail of the protein, recruitment of adapter proteins and intracellular phosphatases, and dampening of signals produced by various kinases.

In order to examine whether the signaling cascade initiated by *Candida* activation of TIGIT is similar to the one described for the endogenous ligands we used TIGIT-expressing YTS cells mutated in the ITIM domain which is critical for the initiation of the signaling cascade. We used two mutants: one with a point mutation in the ITIM's tyrosine residue (YTS TIGIT Y231A), and one with a stop codon interrupting the ITIM and the final 13 amino acids following it (YTS TIGIT Y231Stop). This model was used in the past by our lab and others to characterize the signaling events responsible for the inhibitory effects of TIGIT (Stanietsky, N. *et al.* (2009) 'The interaction of TIGIT with PVR and PVRL2 inhibits human NK cell cytotoxicity.', *Proceedings of the National Academy of Sciences of the United States of America*), (Liu, S. *et al.* (2013) 'Recruitment of Grb2 and SHIP1 by the ITT-like motif of TIGIT suppresses granule polarization and cytotoxicity of NK cells', *Cell Death and Differentiation*). As can be seen in new figure 2e, any interference with the ITIM motif completely abrogated the *Candida*-mediated TIGIT inhibition. We now also discuss this in the results and the discussion sections (see pages 8-9 and 17).

Changes accepted as they are.

8. Also in this regard the Als-TIGIT Interaction may be relevant for mediated NK cell activity. But the effect on T cells needs to be included. In this regard, if the concept of Als-TIGIT interaction is really important than the effect on the host immune response mediated by NK and by T cells needs to be explored

We completely agree with reviewer #2. As mentioned in point #3, we have added both in vitro and in vivo data regarding the effects of the *Candida*-TIGIT interaction (please see new figures 2c-d and 5ab).

Changes accepted as they are.

9. Figure 1. One problem in this figure is presented in panels 1b, 1c, and 1d by the presentation of relative values. What are the real binding data. Like in 1c, a value two times about control may not be a dramatic high affinity binding.

We thank reviewer #2 for this important comment. We also agree that presentation of raw data together with relative values is important in order to report the data in a way that combines and maximizes both transparency and comprehensibility. To that end, we included a histogram showing representative raw data with most experiments (for example Fig. 1a, 3c and 5c).

To provide the reviewer with the requested values we bring below graphs showing the raw data corresponding to figures 1b, 1c and 1d. One of the main reasons we choose to show quantifications of multiple FACS and ELISA experiments using data relative to a constant control is due to high interexperimental changes occurring due to various technical issues such as using different machines, different settings, different experimenters and so on. This can also be observed in the raw data presented below. To present the data accurately we changed the Y-axis from linear to logarithmic in the following graphs (in the FACS experiments). In addition, one specific data point in fig. 1b is significantly different from the rest, so we present the data with and without it (fig. 1b1 vs 1b2).

Reply accepted.

10. For panels 1e and 1f. Whether controls include like using the mAB to block the interaction. How is the effect on binding to the various als knock out mutants. In particular given the redundancy of the various Als proteins this aspect needs to be addressed .

Panels 1e and 1f present an immunofluorescence microscopy experiment in which we stained yeast and hyphae candida cells to understand whether the binding we observed between TIGIT and candida yeast cells is also present in other morphologies of *C. albicans*. Due to that, we only performed controls necessary for the technical aspect of experiment (such as using a control-Ig protein as a negative control and staining yeast cells as a positive control). We agree with reviewer #2 that additional characterization of this interaction, including the specific effects of the different Als ligands or the blocking antibody we use in experiments in later parts of the manuscript are indeed important. We tried using confocal microscopy to characterize and quantify these interactions, but due to the limitation in quantification inherent in the method, we did not succeed, although a trend was observed. Therefore, we prefer showing more reliable quantitative assays, such as FACS, ELISA, BW and NK cytotoxicity (fig. 3a,b,c,d for the effect of the different Als ligands and fig. 2a, 2b, 2d and 3e for the effects of the antibody). If reviewer #2 still feels that the microscopy experiment presented in fig. 1e-f is missing critical controls, we will repeat the confocal experiments many times to obtain statistical differences.

Reply accepted.

11. Figure 2. What is the meaning of the clinical strains. The clinical strains are an impressive selection. It would be nice to compare the expression levels of the various Als isoforms at least in several of the high and low effective isolates.

The mechanism behind the differential ability of the clinical *C. albicans* isolates to activate TIGIT is indeed an interesting question. We checked the Als mRNA expression levels of the five most activating isolates and the three weakest activating isolates using quantitative real-time PCR (please see figure below). As controls we also measured the Als expression levels in the WT lab strain (SC5314) and its three daughter strains deleted for Als6, Als7 or Als9.

As can be seen in the figure, we did not identify any significant differences. This suggests that another mechanism, most likely a post-transcriptional one, is responsible for the difference in TIGIT activation abilities in the examined strains. Such differences can be on the protein level, for example differences in translation efficiencies, protein degradation rates, protein modifications, protein localization, and so on. Another possibility might be that the Als proteins work as a complex and could be regulating the activity and/or expression levels of each other, creating a complex regulatory network between themselves (Hoyer, L. L. and Cota, E. (2016) 'Candida albicans agglutinin-like sequence (Als) family vignettes: A review of als protein structure and function', *Frontiers in Microbiology*).

Unfortunately, to the best of our knowledge no antibodies are available that directly bind Als6, Als7, Als9-1 or Als9-2. As such, probing differences on the protein level is highly complicated. In addition, it is likely that each clinical isolate evolved a different mechanism to change its Als protein activity. We now discuss this in the discussion section (see page 17).

Reply accepted.

12. Figure 3. The effect TIGIT activation for all nine als knock out mutant were assessed. What is the contribution of each als isoform on expression level and on TIGIT mediated activation. Please show also the raw data and not only the relative effects.

The aforementioned experiment is complex. For example, we realized that TIGIT activation levels are affected by the specific growth conditions of the BW cells, and that the live Candida cells affect the BW cells and their ability to be activated and secrete IL-2, probably due to the sensing of fungal metabolites and the secretion of fungal proteases, etc..

Due to that, we analyzed the experiments' results using three stages.

First, we perform the co-incubation of the Candida cells with either parental BW cells or TIGIT-expressing BW cells. BW cell activation is measured by the secretion of IL-2 (using anti-IL-2 ELISA).

Next, we normalize the results from each BW TIGIT x Candida sample to an identical reaction with the same Candida strain in the presence of the BW parental cells. This allows us to better control for the various non-TIGIT related effects and artifacts. Finally, we normalize these results to the measured values of the BW TIGIT x WT Candida in each experiment, to control for inter-experimental variability (which tends to be quite high sometimes). From our experience this experimental scheme provides accurate and better controlled results. Because of the reviewer's comment, we understood that this process isn't explained well enough in the manuscript, so we edited and improved it (please see the relevant methods sub-section, pages 28-29). In addition, we provide below the raw data obtained from the different stages:

Data analysis stages for the BW activation assay. TIGIT activation was assayed using the murine thymoma cell line BW expressing a chimeric TIGIT-z-chain receptor. The BW cells were co-incubated for 48 hours in the

presence or absence of the WT *C. albicans* strain SC5314 or mutant strains deleted for members of the Als protein family. Identical wells containing parental BW cells not expressing TIGIT were also prepared for each experimental condition. TIGIT activation was measured using ELISA for IL-2 secreted by the activated BW cells. Shown are averages of 2-4 independent experiments. The upper figure shows the raw OD values read from the ELISA plates. The middle figure shows these values after normalization to their matched parental BW samples for better control of various internal experiment noise. The lower figure shows the final data after normalization to the WT candida strain in each experiment, in order to better control for noise and technical differences between different experiments.

Reply accepted

13. Figure 4. The effect a control protein (PVR-Ig) is comparable to Als7 protein. Does this reflect a specific interaction and high affinity binding? As negative control and lacking IgG domain needs to be included. In addition, k_{on} and k_{off} values and rates need to be presented. A critical interpretation of this results is that the also Ig fusion proteins show related binding at the human ligand PVR Ig. However, a negative control needs to be presented.

1. Indeed, the MST results for the interaction between TIGIT and both NT-Als7-Ig and PVR-Ig are very similar (40-50nm), with the binding affinity for NT-Als6-Ig even stronger (6nm) and the affinity for NT-Als9-2-Ig weaker (455nm). All these interactions are within an order of magnitude lower than our positive control; the well characterized interaction between TIGIT and its endogenous ligand PVR. As such, we conclude that all these interactions are biologically relevant. Additional support for this conclusion is the fact that the binding affinities between the Als-Ig fusion proteins and TIGIT are similar to those observed in the BW experiment (fig. 3a). Deletion of Als6 leads to the strongest effect on TIGIT activation and deletion of Als9 leads to the weakest. We have also edited the relevant part in the text to better clarify and emphasize these points (see page 12).

2. We agree that a negative control should be included. A negative control was of course included in the original experiment but not shown. This is fixed now, and the new figure 4 includes it.

3. Potential artifacts originating from the IgG Fc domain were controlled by using a negative control protein that includes an Ig tag, and now appears in the new figure 4 and the text (see page 12). In addition, we feel confident that it does not significantly affect the interaction with TIGIT-Ig as we have performed several similar MST experiments with various ligands of TIGIT fused to this tag in different published projects over the years (for example in (Reches, A. *et al.* (2020) 'Nectin4 is a novel TIGIT ligand which combines checkpoint inhibition and tumor specificity', *Journal for Immunotherapy of Cancer*).

4. Unfortunately, k_{on} and k_{off} values are not possible to get when using MST (this is a known limitation of the instrument).

Reply and changes accepted as they are.

14. Figure 5. The effect Figure 5a and 5b: Figure 5d again please present the experimental data without the calculation of the value relative to control. What is the effect in T cell depleted animals.

1. New T cell data was added to new figures 5a and 5b, showing that T cells are also involved the control of invasive candidiasis. In accordance with these results we added further discussion of this subject to various sections of the manuscript (see pages 8-9 and 15-16).

2. We agree with the reviewer that it is important to take into account both the raw data and the relative one to properly assess the reported effect. As mentioned in a previous comment, we initially added the representative histogram (fig. 5c), in addition to the quantification of the relative results (fig. 5d). We think that the combination of the two data formats provides the best balance between data transparency and comprehensibility.

In addition, we show below a quantification of the raw data without normalization to the control-Ig. Please notice that the Y-axis was converted into log scale in order to better present the data. This is due to high variability in the signal strength between the experiments (due to using different FACS machines at different times using slightly different technical settings) and can be an example for our reasoning behind presenting relative data in addition to the histogram presenting the raw data.

Reply accepted.

15. Figure 6. The effect of the various *C. albicans* *als* knock out mutant in the in vivo infection model need to be evaluated and discussed first. Then the role of mAB can be addressed first. What is the reason on selecting got the three mutants, *als6*, *als7* and *als9*. Additional control like revertant strains for each knock out mutant studied need to be presented.

1. We re-ordered the figure and its corresponding parts in the manuscript accordingly, as suggested (see pages 13-14).

2. We chose to select the three *Als* mutants, *als6*^Δ, *als7*^Δ and *als9*^Δ for further in-vivo evaluation as the experiments depicted in figures 3 and 4 indicate that they are the major ligands of TIGIT present on *C. albicans*. We thought that in vivo experiments comparing the effects of TIGIT blockade on the mutants and WT strains will further validate their role as TIGIT ligands under physiological conditions. Another reason behind their inclusion was that they also serve as a control for the possible side effects of TIGIT blockade during invasive candidiasis. It is possible that TIGIT blockade can unspecifically disinhibit the immune response, as seen as a common complication of immune checkpoint inhibition in human patients. If that were to happen in our model, we would have expected to see the same effect during TIGIT blockade in all treated mice, independent of the

candida strain infecting them. On the other hand, if the effects of the anti-TIGIT antibody are specific to the Als immune evasion phenomenon, and are not due to the potential general immunedisinhibitory effect of checkpoint inhibitors, the usage of the ligand-missing mutants would abrogate this effect. We edited the text to better describe these points (see pages 13-15).

3. Adding revertants is indeed a nice suggestion but it requires a very long time to complete these experiments (in our opinion more than a year, as we have to generate them and then test them using the various in vitro and in vivo assays). As we are already working on this project for around five years, we hope that the reviewer will agree to the current manuscript being published without the revertants. We will work on them in the future related projects.

Changes and comments accepted as they are.

16. Language: The text needs revision in language and also in content. Some of the statements sound rather dramatic and in the end are not absolutely right. E.g. The point 'that *Candida albicans* is ... one of the deadliest causes of bloodstream infections' sounds very dramatic but is not completely right (page 2, abstract first sentence). Do the authors really want to say that other pathogenic bacteria like *Streptococcus pneumoniae*, *S. pyogenes*, *Staphylococcus aureus* are not relevant. In particular not to mention the current COVID-19 situation which gives another example on severity of blood stream infections.

We thank the reviewer for this critical point as we always wish to report our data in an accurate and leveled fashion. As such, we went over the text and modified the relevant statements (see page 2 and additional statements throughout the paper).

Regarding the specific point the reviewer raised, we referred to the data presented in the introduction showing that *C. albicans* infection is extremely common, being the 3rd to 4th most common cause of bloodstream infection in hospitalized patients in many countries (and being responsible for 18-22% of such infections in the US). It is also extremely deadly, as 40-60% of all infected patients die (Pappas, P. G. *et al.* (2018) 'Invasive candidiasis', *Nature Reviews Disease Primers*). We did not, by any means, try to imply that other pathogens, whether they be bacteria such as *Staphylococci* or *Streptococci*, viruses such as SARS-COV-2, or other fungi or even parasites, are less relevant.

Unfortunately for patients, all of these pathogens are still equally deadly and important and lead to high morbidity and mortality. The case of COVID-19 which the reviewer brought up is of specific interest, not only due to the global pandemic, but also due to the fact that like most intensive care admitted patients (and especially those treated with steroids), individuals suffering from COVID-19 are at a very high risk of fungal bloodstream infections which takes the lives of many of them. In order to prevent further misunderstandings regarding this statement we amended and softened it, and went over the text to amend similar statements.

Changes accepted as they are.